# Meiotic protein SYCP2 confers resistance to DNA-damaging agents through R-loop-mediated DNA repair

Yumin Wang [1,17], Boya Gao[1,2,17], Luyuan Zhang[3], Xudong Wang[1], Xiaolan Zhu [1], Haibo Yang[1], Fengqi Zhang[1,2], Xueping Zhu[1], Badi Zhou[1], Sean Yao[1], Aiko Nagayama [1,4], Sanghoon Lee [5,6,7], Jian Ouyang[1], Siang-Boon Koh [8], Eric L. Eisenhauer[9,10], Dominique Zarrella[11], Kate Lu[12], Bo R. Rueda [9,10,11], Lee Zou [1,13,14], Xiaofeng A. Su[12], Oladapo Yeku [1,15,16], Leif W. Ellisen [1,4], Xiao-Song Wang [5,6,7] & Li Lan [1,2]✉

Drugs targeting the DNA damage response (DDR) are widely used in cancer therapy, but resistance to these drugs remains a major clinical challenge. Here, we show that SYCP2, a meiotic protein in the synaptonemal complex, is aberrantly and commonly expressed in breast and ovarian cancers and associated with broad resistance to DDR drugs. Mechanistically, SYCP2 enhances the repair of DNA double-strand breaks (DSBs) through transcription-coupled homologous recombination (TC-HR). SYCP2 promotes R-loop formation at DSBs and facilitates RAD51 recruitment independently of BRCA1. SYCP2 loss impairs RAD51 localization, reduces TC-HR, and renders tumors sensitive to PARP and topoisomerase I (TOP1) inhibitors. Furthermore, our studies of two clinical cohorts find that SYCP2 overexpression correlates with breast cancer resistance to antibody-conjugated TOP1 inhibitor and ovarian cancer resistance to platinum treatment. Collectively, our data suggest that SYCP2 confers cancer cell resistance to DNA-damaging agents by stimulating R-loop-mediated DSB repair, offering opportunities to improve DDR therapy.

Breast and ovarian cancers are among the most commonly diagnosed and lethal malignancies globally among women. For most patients with breast and ovarian cancer, chemotherapeutics that induce DNA damage and activate the DNA damage response (DDR), such as cisplatin and carboplatin[1], or drugs targeting DDR, such as Poly (ADP-Ribose) Polymerase inhibitors (PARPi)[2] and Topoisomerase1 inhibitors (TOP1i)[3,4], remain the mainstay of treatment after surgical tumor removal. Collectively, these DDR drugs kill cancer cells by exploiting

[1]Massachusetts General Hospital Cancer Center, Harvard Medical School, 13th Street, Charlestown, MA 02129, USA. [2]Department of Molecular Biology and Microbiology, Duke University School of Medicine, 213 Research Drive, Durham, NC 27710, USA. [3]Emory University School of Medicine, Atlanta, GA 30322, USA. [4]Ludwig Center at Harvard, Boston, MA 02215, USA. [5]UPMC Hillman Cancer Center, University of Pittsburgh, 5117 Centre Ave, Pittsburgh, PA 15232, USA. [6]Department of Pathology, University of Pittsburgh, Pittsburgh, PA 15232, USA. [7]Department of Biomedical Informatics, University of Pittsburgh, Pittsburgh, PA 15232, USA. [8]School of Cellular & Molecular Medicine, University of Bristol; University Walk, Bristol BS8 1TD, UK. [9]Division of Gynecologic Oncology, Department of Obstetrics and Gynecology, 55 Fruit St, Massachusetts General Hospital, Boston, MA 02114, USA. [10]Obstetrics, Gynecology and Reproductive Biology, Harvard Medical School, Boston, MA 02115, USA. [11]Vincent Center for Reproductive Biology, Department of Obstetrics and Gynecology, 55 Fruit St, Massachusetts General Hospital, Boston, MA 02114, USA. [12]David H. Koch Institute for Integrative Cancer Research, Department of Biology, Massachusetts Institute of Technology, Cambridge, MA 02139, USA. [13]Department of Pathology, Massachusetts General Hospital, Harvard Medical School, 55 Fruit St, Boston, MA 02114, USA. [14]Department of Pharmacology & Cancer Biology, Duke University School of Medicine, 213 Research Drive, Durham, NC 27710, USA. [15]Division of Hematology-Oncology, Massachusetts General Hospital, 55 Fruit St, Boston, MA 02114, USA. [16]Department of Medicine, Massachusetts General Hospital, 55 Fruit St, Boston, MA 02114, USA. [17]These authors contributed equally: Yumin Wang, Boya Gao. ✉e-mail: li.lan@duke.edu

their DNA repair defects or genomic instability, which are prevalent in breast, ovarian, and some other types of cancers. Many DDR drugs directly or indirectly induce DNA double-strand breaks (DSBs), a lethal form of DNA damage. DSB repair is especially important for cancer cell survival in response to DDR drugs. Despite the efficacy of DDR drugs in the clinic, drug resistance is inevitable in the majority of patients with advanced disease[5]. Thus, it is important to understand the mechanisms underlying DDR drug resistance and develop strategies to predict and overcome the resistance for improving the treatment of cancer patients.

Homologous Recombination (HR) is a pathway that is crucial for the repair of DSBs. It is well known that breast and ovarian cancer cells with *BRCA1/2* mutations are defective in HR and highly sensitive to PARPi[5–8]. However, *BRCA1/2* mutations are only detected in a small fraction of breast and ovarian cancer patients around 10%, limiting the effective use of PARPi and other DDR drugs in the clinic. Therefore, a better understanding of the mechanisms of repair is critical for exploiting the genomic vulnerabilities of breast and ovarian cancers. In HR-deficient cancer cells, restoration of the canonical HR and upregulation of HR and/or alternative repair mechanisms have been reported to confer DDR drug resistance[9]. In cancers, transcription-coupled HR (TC-HR) is shown to contribute to cancer cell survival because cancer cells consistently deal with increased levels of damage and are "transcription addicted"[10–15]. In TC-HR, RNA transcripts form R-loops by hybridizing with DNA to facilitate repair[12]. Furthermore, the R-loops induced by DNA damage promote RAD51-dependent HR, contributing to the DNA damage resistance of cancer cells[10,12–14,16,17]. Therefore, proteins involved in R-loop formation and R-loop-dependent TC-HR could serve as biomarkers or targets in therapy to predict and overcome DDR drug resistance.

Here, we discovered Synaptonemal Complex Protein 2 (SYCP2) as a regulator of R-loops and TC-HR. SYCP2 is a component of the Synaptonemal complex (SYC), which functions in the prophase of meiosis to pair homologous chromosomes[18,19]. A limited number of correlative studies has shown aberrant SYCP2 expression in breast and papillomavirus-positive oropharyngeal carcinoma[20,21], while mechanisms, potential clinical, and therapeutic relevance of SYCP2 in cancers have not been well defined. Here, we show that SYCP2 expression is aberrantly upregulated in breast cancer, ovarian cancer, and certain other cancer types. In contrast, SYCP2 is not expressed in normal tissues except in the testis. Importantly, the aberrant expression of SYCP2 in cancer cells strongly correlates with the resistance to a broad spectrum of DDR drugs, but not drugs targeting other signaling pathways. Our biochemical and cell biological studies reveal that SYCP2 promotes R-loop formation and RAD51 localization to DSBs through its lysine (K) and arginine (R) enriched motif, thereby stimulating TC-HR. SYCP2 overexpression enhances TC-HR and leads to resistance to PARPi and other DDR drugs independently of BRCA1 or BRCA2. Conversely, SYCP2 depletion impairs TC-HR and promotes increasing sensitivity to PARPi and other DDR drugs. We show that in breast and ovarian cancer patients, high SYCP2 expression is associated with poor prognosis and resistance to an antibody-conjugated TOP1 inhibitor and platinum, respectively. Together, our results suggest that aberrant expression of SYCP2 in cancer promotes TC-HR and confers DDR drug resistance, revealing a promising biomarker to predict the resistance to DDR drugs and a potential target overcome in cancer therapy in the future.

## Results

### Aberrant expression of the synaptonemal complex protein 2 (SYCP2) in cancer

To identify genes whose up-regulation is associated with the resistance to drugs targeting DDR, we designed a multi-step bioinformatics pipeline to analyze gene expression in breast cancer cell lines. First, we selected 49 breast cancer cell lines from the Cancer Cell Line Encyclopedia (CCLE) and Genomics of Drug Sensitivity in Cancer (GDSC) databases. These cell lines were selected because they have been characterized for both gene expression and sensitivities to DDR drugs, including PARPi, TOP1i, and cisplatin. Next, we performed a correlation analysis between the half-maximal inhibitory concentration (IC50) of each of these DDR drugs and the expression of genes in the 49 cell lines. The expression of 152 genes was positively correlated with the IC50s of the three DDR drugs (R-value > 0.4) (Fig. 1A, Supplementary Fig. 1A). Quantifications of gene expression changes in breast and/or ovarian cancers compared to normal tissue revealed that expression of 7 of the 152 genes was increased more than 5-fold in cancer (Fig. 1A). Among the 7 genes, *GATA3*, *FSIP*, and *FOXA1* have been suggested to associate with chemoresistance in ER-positive breast cancers[16,22]. Notably, in both Cancer Genomic Atlas (TCGA) and CCLE databases, the meiotic gene *SYCP2* is commonly upregulated in breast, cervical, and ovarian cancers and also detected in other cancer types, including lung, head-and-neck, and bladder cancers (Fig. 1B). Immunohistochemistry (IHC) analysis of breast cancer tissues and adjacent normal breast tissues from patients showed that SYCP2 protein was specifically detected in tumors but not in adjacent normal tissues (Fig. 1C). Moreover, SYCP2 was detected in an ovarian tumor sample from a patient but not in a normal ovary by IHC (Supplementary Fig. 1B). SYCP2 is known to be present in normal tissues except testis, where SYCP2 functions in meiosis[18,19]. In the TCGA database, SYCP2 expression is significantly higher in breast tumors than in adjacent breast tissue (Fig. 1D). We found that DNA hypomethylation at two sites near the SYCP2 promoter strongly correlated with high SYCP2 expression in 309 breast cancer samples (Fig. 1E). In addition, DNA hypomethylation at these two sites was only detected in breast tumors but not in adjacent normal breast tissues (Supplementary Fig. 1C). Therefore, our results suggest the meiotic gene SYCP2 is aberrantly expressed in cancer due to DNA hypomethylation. Importantly, among the SYC family genes, SYCP2 is the only gene that showed significant upregulation in breast cancer compared to normal breast samples (Supplementary Fig. 1D) according to the TCGA database.

### SYCP2 expression is correlated with DDR drug resistance

To further examine the expression of SYCP2 in cancer, we stained SYCP2 by IHC in tumor samples from breast cancer patients. In parallel with the IHC analysis, RT-PCR was performed to quantify the levels of SYCP2 mRNA in tumors and normal tissues. SYCP2 protein and mRNA levels are both upregulated in tumors compared to normal tissues. The increases of SYCP2 protein and mRNA are overall correlated, confirming that SYCP2 is not only transcriptionally upregulated but also expressed as a stable protein in tumors (Fig. 1F). In breast and ovarian cancer cell lines, high SYCP2 expression correlated with resistance to Cisplatin, a DNA crosslinking agent, and Camptothecin 11 (CPT11, irinotecan), an inhibitor of TOP1 (Fig. 1G). To test whether SYCP2 expression also correlates with the resistance to drugs targeting other pathways, we expanded the correlation analysis to 248 anticancer drugs affecting different cellular pathways. Among all the pathways targeted by drugs, the resistance to drugs targeting the DDR pathway showed the strongest positive correlation with the expression of SYCP2 in several cancer types including breast and ovarian cancers (Supplementary Fig. 1E). In contrast, drugs targeting PI3K kinase and histone-modifying enzymes show negative correlation with SYCP2 expression (Supplementary Fig. 1E). Together, these results suggest that the upregulation of SYCP2 in cancer preferentially affects the response to DDR-targeted drugs.

Furthermore, SYCP2 expression correlates with resistance to three distinct PARP inhibitors (PARPis): Olaparib, Talazoparib, and Rucaparib (Fig. 1H). The correlation coefficient r values for SYCP2 expression and IC50s of Talazoparib, Rucaparib, and Olaparib are 0.402, 0.509, and 0.627, respectively. Moreover, the *p*-values for the positive correlations of SYCP2 expression with IC50s of PARPis are in

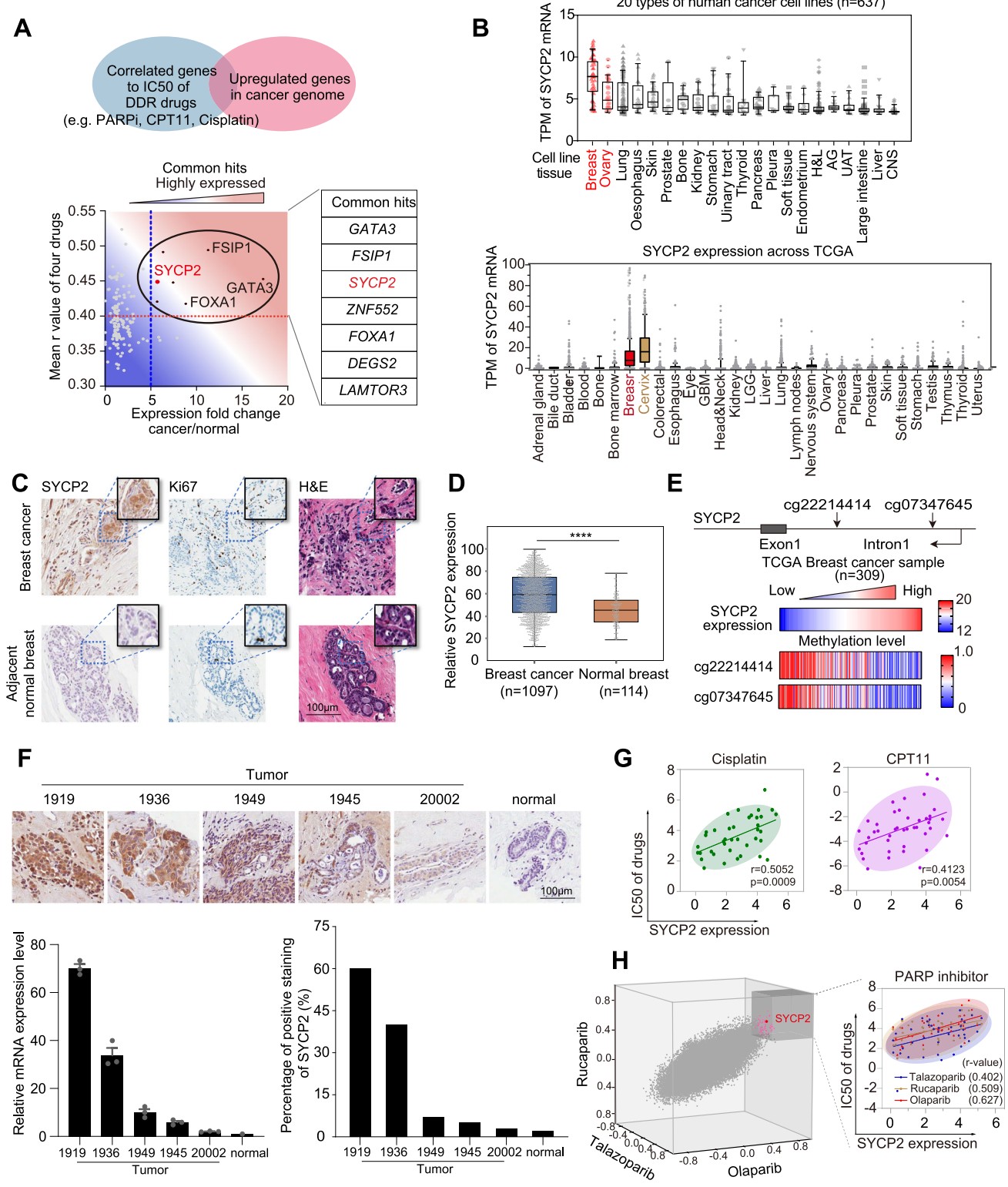

the range of $1\times10^{-6}$–0.019, suggesting that high SYCP2 expression may be a potential predictor of resistance to PARPi and other DDR drugs.

## SYCP2 promotes homologous recombination (HR)

Given that many DDR drugs directly or indirectly induce DSBs and SYCP2 expression in cancer correlates with resistance to DDR drugs, we tested whether SYCP2 is involved in DSB repair. The cervical cancer cell line HeLa and osteosarcoma cell line U2OS are widely used in the

studies of DSB repair, and both these cancer cell lines express SYCP2. Knockdown (KD) of SYCP2 in cervical cancer cell line HeLa and a panel of breast cancer cells by siRNA (siSYCP2) significantly reduced colony forming rate of these cancer cell lines, however, effects of SYCP2 in normal BJ and normal breast MCF10A cells were subtle (Supplementary Fig. 2A), suggesting that SYCP2 contributes to the fitness of cancer cells. SYCP2 KD sensitized cells to PARPi and Cisplatin in HeLa cells (Fig. 2A) and U2OS cells (Supplementary Fig. 2B). Upon ionizing radiation (IR), SYCP2 KD significantly delayed the clearance of γ-H2AX

**Fig. 1 | SYCP2 expression in cancer associates with resistance to drugs targeting DDR pathways. A** Common hits of genes which are highly expressed in cancer and their high expression correlates with DDR drugs' resistance. The X-axis indicates the median value expression of the correlated genes in breast cancer over normal. The Y-axis indicates the mean correlation r-value between IC50 to DDR drugs [Olaparib, a PARPi; CPT11, a TOP1i, Cisplatin, a crosslinking agent]. The comparison for the analysis is one-sided. *r*-values > 0.4 with significant p values by Pearson correlated analysis were used. **B** Upregulation of expression of SYCP2 in indicated types of tumors from CCLE and TCGA database, respectively. TPM was used for the normalization of gene expression. For box-plot, the minimum to maximum are shown with all data points labeled. **C** IHC of SYCP2, Ki67, and H&E staining in breast cancer and adjacent normal breast tissues (MGH patient #20006). Enlarged images are shown in upper right. **D** Expression of SYCP2 in breast cancer compared to normal breast tissues in TCGA database. For box-plot, the minimum to maximum are shown with all data points labeled. **E** Upper panel: Scheme of two loci (cg22214414 and cg07347645) of SYCP2 intron 1. The heatmap showed high SYCP2 expression correlates with low methylation at loci cg22214414 and cg07347645. The data was collected from TCGA (*n* = 309 samples). **F** The representative images of IHC staining of SYCP2 in breast cancer and para cancer tissues from breast cancer patients were shown on the top. The relative RNA expression levels and relative protein levels from tissues were quantified by qRT-PCR and IHC, respectively. SYCP2 mRNA from qRT-PCR were normalized to GAPDH (*n* = 3 experiments, Mean +/− SEM); the percentage of positive IHC staining of SYCP2 in tumor tissues were quantified. **G** The two-dimensional (2D) plot of correlation between SYCP2 expression and IC50 of Cisplatin and CPT11 in breast cancer cell lines. R-values are 0.51 and 0.41, respectively. **H** 3D and 2D plot of correlation between SYCP2 expression and IC50 of Olaparib, Rucaparib, and Telozoparib. *R*-values are 0.63, 0.51, and 0.40, respectively. Source data are provided as a Source Data file.

foci after 24 hours (Fig. 2B & Supplementary Fig. 2C). These results suggest that SYCP2 is required for efficient DSB repair in cancer cells. Since HR deficiency contributes to PARPi and cisplatin sensitivities[5–8], we tested the effects of SYCP2 KD and overexpression (OE) on HR efficiency. Using the DR-GFP reporter for HR activity in U2OS cells[23], we found that SYCP2 KD reduced HR, whereas SYCP2 OE enhanced HR (Fig. 2C). In a normal breast MCF10A cell line, SYCP2 OE enhanced HR as well (Fig. 2C right). It should be noted that SYCP2 KD did not significantly alter the cell cycle at the time when HR was affected (Supplementary Fig. 2D). We also performed experiments to assess the impact of SYCP2 knockdown and overexpression on the repair frequency of non-homologous end joining (NHEJ), alternative non-homologous end joining (alt-NHEJ), or single strand annealing (SSA) pathways, using EJ5-GFP, EJ2-GFP, SSA-GFP reporter assay, respectively[24] (Supplementary Fig. 2E). Repair efficiencies in above reporter assays upon knocking down or overexpressing (OE) SYCP2 were not changed significantly. These results indicate that SYCP2 does not have a substantial influence on the occurrence of these DNA repair pathways. Thus, SYCP2 may preferentially contribute to HR activity especially in cancer cells expressing SYCP2.

Next, we sought to understand how SYCP2 functions in HR. DNA end resection by the MRE11-RAD50-NBS1 complex is an early event in HR that generates ssDNA, which recruits RPA and then RAD51 to DSBs[25]. To investigate the resection efficiency, we utilized the AID-DlvA reporter system[26]. Remarkably, SYCP2 knockdown did not reduce end resection efficiency at DSBs, whereas MRE11 knockdown led to a significant decrease in the resection rate (Supplementary Fig. 2F). Moreover, number of ionizing radiation-induced foci (IRIF) of CtIP and phosphorylated RPA (pRPA) was not affected by siSYCP2 (Supplementary Fig. 2G). Knocking down end resection enzymes, e.g. Mre11 and CtIP did not affect IRIF of SYCP2 (Supplementary Fig. 2H), suggesting SYCP2 functions in repair of DSBs independently of end resection.

In U2OS cells, SYCP2 was recruited to laser micro-irradiation induced DSBs, and the kinetics of SYCP2 accumulation at DSBs was slower than NBS1 but faster than RPA (Fig. 2D), suggesting SYCP2 is unlikely a direct sensor of DSBs but may be involved in HR prior to RAD51 recruitment. During the meiosis, SYCP2 is required for the formation of synaptonemal complexes between homologous chromosomes, which promotes the subsequent pairing/recombination by RAD51 and its meiotic homologue DMC1[27], raising the possibility that SYCP2 is a regulator of RAD51. IRIF of RAD51 reflect the recruitment of RAD51 to DSBs after DNA end resection[28]. SYCP2 KD did not trigger RAD51 IRIF before damage (Supplementary Fig. 2I), while it significantly decreased RAD51 IRIF without affecting RAD51 expression (Fig. 2E). These results suggest that SYCP2 may facilitate HR by promoting RAD51 foci formation, thereby allowing cancer cell survival.

We further validated the effects of SYCP2 KD in the triple-negative breast cancer cell line MDA-MB-231. As shown in Supplementary Fig. 3A&3B, SYCP2 KD diminished RAD51 IRIF and delayed the

clearance of γ-H2AX foci after IR. We confirmed that siSYCP2 repressed IRIF of Rad51 in Cyclin A-positive marked S/G2 cells in both U2OS and MDA-MB-231 cells (Supplementary Fig. 3B). Consistently, SYCP2 OE increased IRIF of RAD51 (Supplementary Fig. 3C). We also observed clear and comparable IRIF of RPA in both SYCP2 OE and vehicle control groups, indicating that SYCP2 OE did not significantly affect the formation of RPA foci (Supplementary Fig. 3D). In addition to IR, we tested the effects of IRIF of RPA and RAD51 upon other damage agents e.g. PARPi, TOP1i. Following PARPi/TOP1i treatment in U2OS cells, SYCP2 KD resulted in a reduction of RAD51 foci, while having no discernible impact on RPA foci (Supplementary Fig. 3E). Together, these results suggest that, in SYCP2-expressing cancer cells, SYCP2 acts upstream of RAD51 to facilitate RAD51 foci formation upon DNA damage in the HR pathway.

## SYPC2 regulates HR in a transcription-dependent manner

We recently showed that RNA transcripts significantly stimulate HR by forming R-loops at the transcribed region of the genome[12]. Using the Tet-DR-GFP HR assay, a modified DR-GFP HR assay, we can compare the HR activities in transcriptionally on and off states[12]. As we reported, HR efficacy was about 2-fold higher when transcription is on in cells[12] (Fig. 3A black bar). Interestingly, SYCP2 KD reduced HR activity when transcription was on, but did not further reduce HR when transcription was off (Fig. 3A), suggesting that SYCP2 promotes HR in a transcription-dependent manner.

To verify the role of SYCP2 in TC-HR, we next used the DNA damage at RNA transcription sites (DART) assay to monitor the localization of SYCP2 to a site of localized ROS-induced DNA damage[10,13]. In the DART assay, KillerRed (KR), a light excitable ROS-releasing protein, is fused to VP16 transcription activator (TA). When the TA-KR fusion protein is targeted to an array of Tet Response Elements (TRE) at a reporter gene in the genome, it activates transcription and generates local DSBs after light activation. When DNA damage was induced by TA-KR at the locus in the presence of transcription, SYPC2 was preferentially recruited to this TA-KR marked locus (Fig. 3B). When KR was fused to the Tet repressor (tetR-KR), DNA damage was induced at the locus without active transcription. In this situation, tetR-KR generated DNA damage was not sufficient to recruit SYCP2 in the absence of transcription (Fig. 3B). Cherry is a non-damage control for KR. SYCP2 was not recruited to the locus by TA-cherry or tetR-cherry without DNA damage regardless of transcription status (Fig. 3B). In addition to KR induced DSBs, SYCP2 was also recruited to I-SCEI induced DSBs in transcribed regions of the genome (Fig. 3C). Thus, in the DART assay, SYPC2 is recruited to sites of ROS-induced DNA damage in a transcription-dependent manner.

In the DART assay, RAD51 was recruited to the TA-KR bound locus in a transcription-dependent manner[10]. SYCP2 KD significantly reduced RAD51 foci at sites marked by TA-KR (Fig. 3D). We previously showed that R-loops are induced at sites of TA-KR to facilitate RAD51-mediated repair[10]. The R-loops detected at sites of TA-KR by the S9.6

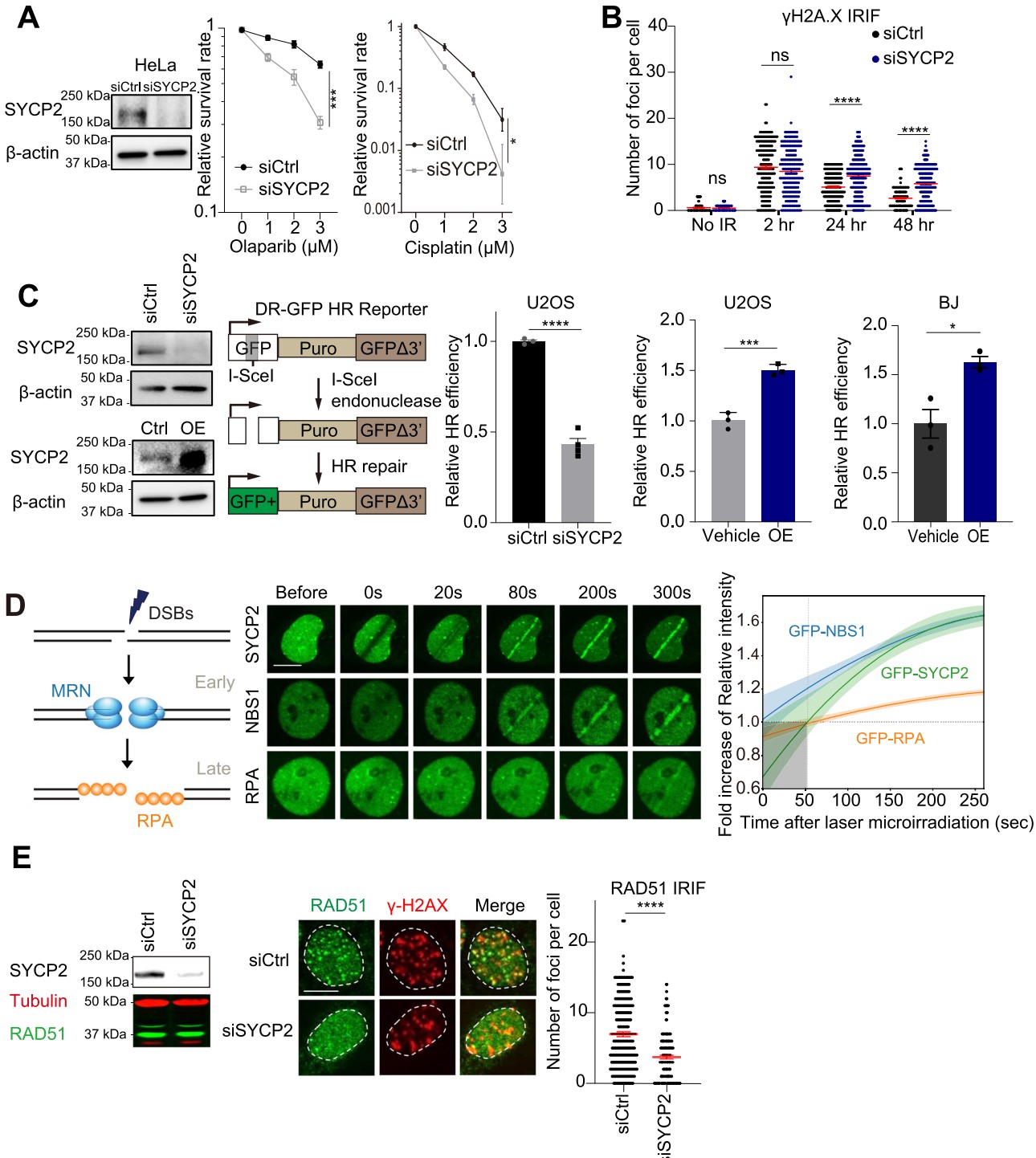

**Fig. 2 | SYCP2 is involved in DDR and is required for cell survival. A** Cell survival rate of HeLa cells with Olaparib and Cisplatin at the indicated dose via colony-forming assay. Three independent experiments were done (*n* = 3 experiments, Mean +/− SEM.). WB of siControl (siCtrl) and siSYCP2 in HeLa was shown. (*p* = 0.0245). **B** Quantification of ionizing radiation-induced foci (IRIF) of γH2AX after 2 Gy IR at indicated hours (hr) of recovery time (*n* = 200 cells, Mean +/− SEM) in HeLa cells. **C** WBs of SYCP2 KD and OE in U2OS cells was shown. Relative HR frequency in siCtrl or siSYCP2 (left), empty vector or SYCP2 overexpression (OE) in U2OS cells (middle), empty vector or SYCP2 OE in BJ cells (right) in the DR-GFP reporter assay were quantified. Three independent experiments were done (*n* = 3

experiments, Mean +/− SEM). (*p* < 0.0001, *p* = 0.003, *p* = 0.0160). **D** Kinetics and representative images of GFP-NBS1, -RPA, and -SYCP2 recruitment in U2OS cells from 0-300 seconds (s) after laser microirradiation. The fold increase of mean intensity (sites of irradiation/nucleus background) was quantified (*n* = 10 cells, Mean +/−SEM). Scale Bar = 10 µm. **E** The numbers of RAD51 IRIF in siCtrl and siSYCP2 treated U2OS cells 1 h after 2 Gy IR were quantified (*n* = 200 cells, Mean +/− SEM). WB of SYCP2 and RAD51 and the representative images of RAD51 and γH2AX IRIF were shown on the left. Scale Bar = 10 µm Statistical analysis was done with the unpaired two-tailed Student-t-test, *****p* < 0.0001. Source data are provided as a Source Data file.

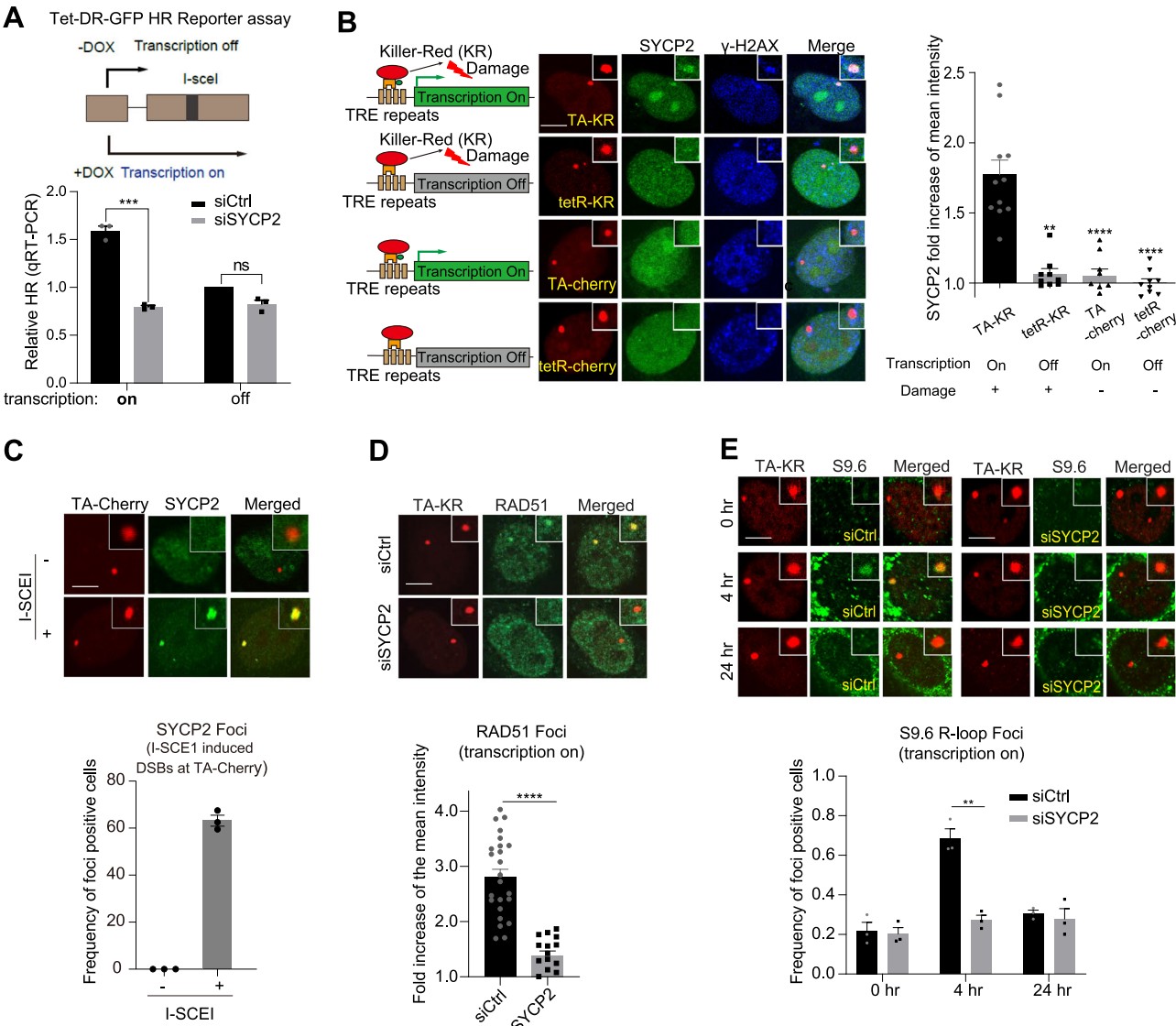

**Fig. 3 | SYCP2 plays an essential role in transcription-coupled homologous recombination (TC-HR). A** Scheme of tet-DR-GFP HR reporter for measuring TC-HR efficiency was shown on top. Relative HR frequency was measured when the transcription is on or off by qRT-PCR in siCtrl or siSYCP2 treated cells. Mean frequency of the HR (qPCR) compared to the siCtrl group at transcription off is shown. Three independent experiments were done (n = 3 experiments, Mean +/− SEM). (p = 0.0002). **B** Scheme of damage induction via KillerRed (KR) at transcribed or non-transcribed region of the genome in the DART assay (left). U2OS-TRE cells were transfected with GFP-SYCP2 and TA-KR/tetRKR/TA-cherry/tetR-cherry. For light activation of KR in all below experiments in the DART assay, cells were light-activated for 20 min and recovered for 30 min. γH2AX foci were stained and are positive at sites of KR but not cherry. SYCP2 is preferentially recruited to sites of TA-KR. Representative images and quantification of recruitment of SYCP2 at the indicated site were shown. Mean intensity of SYCP2 at TA-KR /mean intensity of background was shown (n = 10 cells, Mean +/− SEM). Experiments were repeated 3 times. **C** SYCP2 foci frequency at I-SCEI endonuclease-induced damage sites marked by TA-Cherry in U2OS-TRE cells treated 24 h with or without I-SCEI transfection. Three independent experiments were done (n = 3 experiments, Mean +/− SEM). **D** U2OS-TRE cells transfected with TA-KR and siCtrl/siSYCP2 were light-activated, recovered for 30 min, fixed, and stained with anti-RAD51. Fold increase of RAD51 foci at sites of KR compared to background was quantified (n = 25 cells for siCtrl and n = 16 cells for siSYCP2, Mean +/− SEM). Experiments were repeated 3 times. **E** U2OS-TRE cells transfected with TA-KR and siCtrl/siSYCP2 with or without light-activation were recovered at 4 h and 24 h, then fixed and stained with anti-S9.6. Frequency of S9.6 foci positive cells at TA-KR was counted. Three experiments were done (n = 3 experiments, Mean +/− SEM), 100–200 individual cells were quantified per group. Statistical analysis was done with the unpaired two-tailed Student-t-test, ****p < 0.0001. Scale Bar = 10 μm. Source data are provided as a Source Data file.

antibody, which recognizes DNA-RNA hybrids, are sensitive to RNaseH treatment[10]. Interestingly, SYCP2 OE-mediated RAD51 IRIF were also abolished by the treatment of RNA Polymerease II inhibitors or siRNase H1 (Supplementary Fig. 3C). These results together support the notion that SYCP2 promotes R-loop-dependent HR. Furthermore, the kinetics of R-loop accumulation at sites of TA-KR correlates with repair kinetics[10,14]. As shown in Fig. 3D, the levels of DNA damage-induced R-loops increased to the maximum level around 4 hours (hr) after DNA damage induction and returned to the basal level 24 h after damage

(Fig. 3E). R-loops were not detected when transcription is off in the absence or presence of SYPC2 and as shown in Supplementary Fig. 3F. However, in SYCP2 KD cells, the levels of R-loops were significantly reduced 4 h after DNA damage (Fig. 3E), suggesting that SYPC2 might regulate damage-induced R-loop accumulation. Additionally, the level of R-loops before and after damage at sites of TA-KR remained unchanged regardless of the presence or absence of CtIP, suggesting that function of SYCP2 for regulating R-loops is independent from end resection of DSBs. (Supplementary Fig. 3G)

## SYCP2 binds DNA-RNA hybrids and facilitates R-loop formation

To further investigate the impact of transcription on the recruitment of SYCP2, we examined the effect of RNA polymerase II activity on the recruitment of SYCP2. Cells were treated with the RNA polymerase II inhibitor α-amanitin or DRB, However, we observed that neither DRB nor α-amanitin affected the recruitment of SYCP2 to the TA-KR sites (Supplementary Fig. 4A). Moreover, siRNA against CtIP and Mre11 did not affect both SYCP2 IRIF (Supplementary Fig. 2H) and SYCP2 foci at TA-KR sites (Supplementary Fig. 4B). In addition, inhibitors of DNA damage sensors ATM, ATR, and DNAPK did not show significant effects on SYCP2 recruitment (Supplementary Fig. 4C). SYCP2 is a large protein composed of 1530 amino acids (a.a.), we divided SYCP2 into several fragments to investigate how SYCP2 might regulate R-loops and TC-HR. The N-terminal domain of SYCP2 is conserved in SYCP2L, the C-terminal coiled-coil domain interacts with SYCP3 and the function of the middle (M) domain of SYCP2 remains unknown[19,27,28]. We generated a set of SYCP2 fragments, including M1 (492−1035 a.a.), M2 (1035−1364 a.a.), and ΔM1 (Δ492−1035 a.a.), and tested their damage response capacity in the DART assay. Only the M1 fragment of SYCP2 was efficiently recruited to sites of TA-KR. In contrast, M2 and ΔM1 did not respond to DNA damage efficiently (Fig. 4A, B), indicating that SYCP2L and SYCP3 are not required for damage response of SYCP2. Furthermore, the M1 region of SYCP2 is both necessary and sufficient for the localization of SYCP2 to transcriptionally active DNA damage sites.

Next, we tested whether the SYCP2 fragments are functional in HR. We first used the CRISPR-based mClover HR assay to verify the contribution of SYCP2 to HR at the endogenous *Lamin A* (*LMNA*) gene, which is actively transcribed. A DSB is generated in the *LMNA* gene using CRISPR-Cas9, and a *mCloveer* cassette is inserted into the DSB through HR, resulting in mClover-tagged Lamin A[29]. Similar to that in the DR-GFP HR assay, overexpression of full-length (FL) SYCP2 enhanced HR efficiency in the mClover HR assay (Fig. 4C). Among the SYCP2 fragments, only the M1 fragment substantially increased HR efficiency when overexpressed (Fig. 4C), suggesting that the M1 region of SYCP2 is important for its pro-HR activity.

Knowing the M1 region is important for HR (Fig. 4C), we purified the M1 fragment of SYCP2 and tested its ability to bind various DNA and RNA substrates (Fig. 4D). The binding affinities between SYCP2-M1 with DNA or RNA were quantified using Microscale thermophoresis (MST)[30]. SYCP2-M1 showed a high affinity to DNA-RNA hybrids with a binding affinity (measured the equilibrium dissociation constant/Kd) of 71.5 nM (Fig. 4D). The Kd of SYCP2-M1 for dsDNA and ssRNA are 291 and 535 nM, respectively. These results suggest that SYCP2-M1 binds DNARNA hybrids more efficiently than dsDNA and ssRNA in vitro. Next, we used the electrophoresis mobility shift assays (EMSA) assay to confirm the binding of SYCP2-M1 to DNA-RNA hybrids. We observed a gradual mobility shift of DNA-RNA hybrids in the presence of increasing concentrations of SYPC2-M1 protein (Fig. 4E). These results support the idea that SYCP2 interacts with R-loops at DNA damage sites through the M1 region.

The binding of the M1 fragment to DNA-RNA hybrids prompted us to test whether SYCP2-M1 promotes R-loop accumulation. In the R-loop formation assay, a labeled ssRNA oligo is incubated with a dsDNA plasmid containing homologous sequences in the presence of proteins to be tested[12]. RAD51AP1 is reported to promote R-loop formation in vitro and used as a positive control in the R-loop formation assay[12]. Similar to RAD51AP1, SYCP2-M1 promoted R-loop formation in vitro (Fig. 4F), although less efficiently. The combination of SYCP2-M1 and RAD51AP1 did not significantly stimulate R-loop formation compared to RAD51AP1 alone. Together, these results show that the M1 region of SYCP2 has the ability to promote R-loop formation independently of RAD51AP1, providing a possible explanation of its contribution to damage induced R-loop accumulation in cells.

## A lysine(K)/arginine(R)-rich motif in M1 is required for R-loop formation and TC-HR

The M1 fragment contains a lysine/arginine (KR)-rich region, which could potentially be involved in binding negatively charged polynucleotides. To understand if these KR residues of SYCP2 could be functionally important for damage recruitment and R-loop binding, we created several KR mutants of SYCP2-M1: the SYCP2 M1-11KR mutant contains 11 K/R-to-Alanine (A) mutations in the region of 810−826 a.a., the SYCP2 M1-5KR mutant contains 5 K/R-to-A mutations at 822−826 a.a., and the SYCP2 M1-2KR mutant contains 2 K/R-to-A mutations at 810−811 a.a. (Fig. 5A). OE of M1 and M1-2KR increased HR in the mClover HR assay, but M1-5KR and M1-11KR did not (Fig. 5B, Supplementary Fig. 5A). Thus, the 5 K/R residues at 822−826 a.a. are functionally important for the role of SYCP2 in HR.

To further characterize SYCP2-M1 and derivative mutants in the absence of endogenous SYCP2, we designed the second siSYCP2[#2] that targets a sequence upstream of the M1 region (Fig. 5C). In cells depleted of endogenous SYCP2, exogenously expressed M1 and M1-2KR localized to sites of TA-KR-induced DNA damage efficiently, whereas M1-5KR and M1-11KR did not (Fig. 5D, Supplementary Fig. 5B), showing that the recruitment of SYCP2-M1 and M1-2KR is not dependent on endogenous SYCP2 and the defects of M1-5KR and M1-11KR are not attributed to their inability to interact with endogenous SYCP2. It is worth noting that there is no recruitment of endogenous SYCP2 and RAD51 when transcription is off (Supplementary Fig. 5B right panel & 5C lower panel). SYCP2-M1-5KR and M1-11KR did not rescue RAD51 foci at sites of TA-KR in the absence of endogenous SYCP2 (Fig. 5E, Supplementary Fig. 5C). Expression of M1 and M1-2KR, but not M1-5KR and M1-11KR, restored HR in the LMNA-HR assay when endogenous SYCP2 was knocked down (Fig. 5F).

Using the Tet-DR-GFP HR assay (Fig. 3A), we found M1 but not M1-5KR rescued TC-HR in the absence of endogenous SYCP2 when transcription was on (Fig. 5G). Importantly, the levels of DNA damage-induced R-loops in cells lacking endogenous SYCP2 were rescued by M1 but not M1-5KR (Fig. 5H, Supplementary Fig. 5D). We further conducted EMSA to test the direct binding between SYCP2-M1 to DNA-RNA hybrids. We also found that M1-2KR but not M1-5KR has the affinity with hybrids as compared to the M1 domain (Supplementary Fig. 5E). Together, these results suggest that the 5KR motif (822-826 a.a.) is important for the ability of SYCP2 to bind promote R-loop and promote R-loop-dependent TC-HR.

## SYCP2 has an ability to promote repair independently of BRCA1

BRCA1 and BRCA2 (BRCA1/2) are important for the recruitment of RAD51 in the canonical HR pathway. However, BRCA1/2-independent RAD51 recruitment was also observed. For example, in BRCA2-deficient cancer cells, damage-induced RAD51 foci formed in a RAD52-dependent manner[31]. An RNF168 and PALB2-mediated mechanism of RAD51 loading was observed in BRCA1-dfeicient cell[32]. Furthermore, when ROS-induced DNA damage occurs in transcribed regions, RAD51 is recruited in an R-loop and RAD52-dependent but BRCA1/2-independent manner[10]. The role of SYCP2 in recruiting RAD51 in the DART assay raised the possibility that SYCP2 has the ability to function independently of BRCA1/2.

To investigate the functional relationship between SYCP2 and BRCA1, we first tested whether the correlation between *SYCP2* expression and DDR drug resistance is affected by the BRCA1 status. The correlation between *SYCP2* expression and DDR drug resistance is similar in cell lines with or without *BRCA1* mutations, or in cell lines expressing *BRCA1/2* at high or low levels (Fig. 6A), suggesting that *SYCP2* expression affects DSB repair largely independently of the *BRCA1/2* status. Consistent with this, BRCA1 KD affected neither SYCP2 expression nor SYCP2 localization to sites of TA-KR induced damage (Fig. 6B, Supplementary Fig. 6A). BRCA1 and SYCP2 also formed IRIF

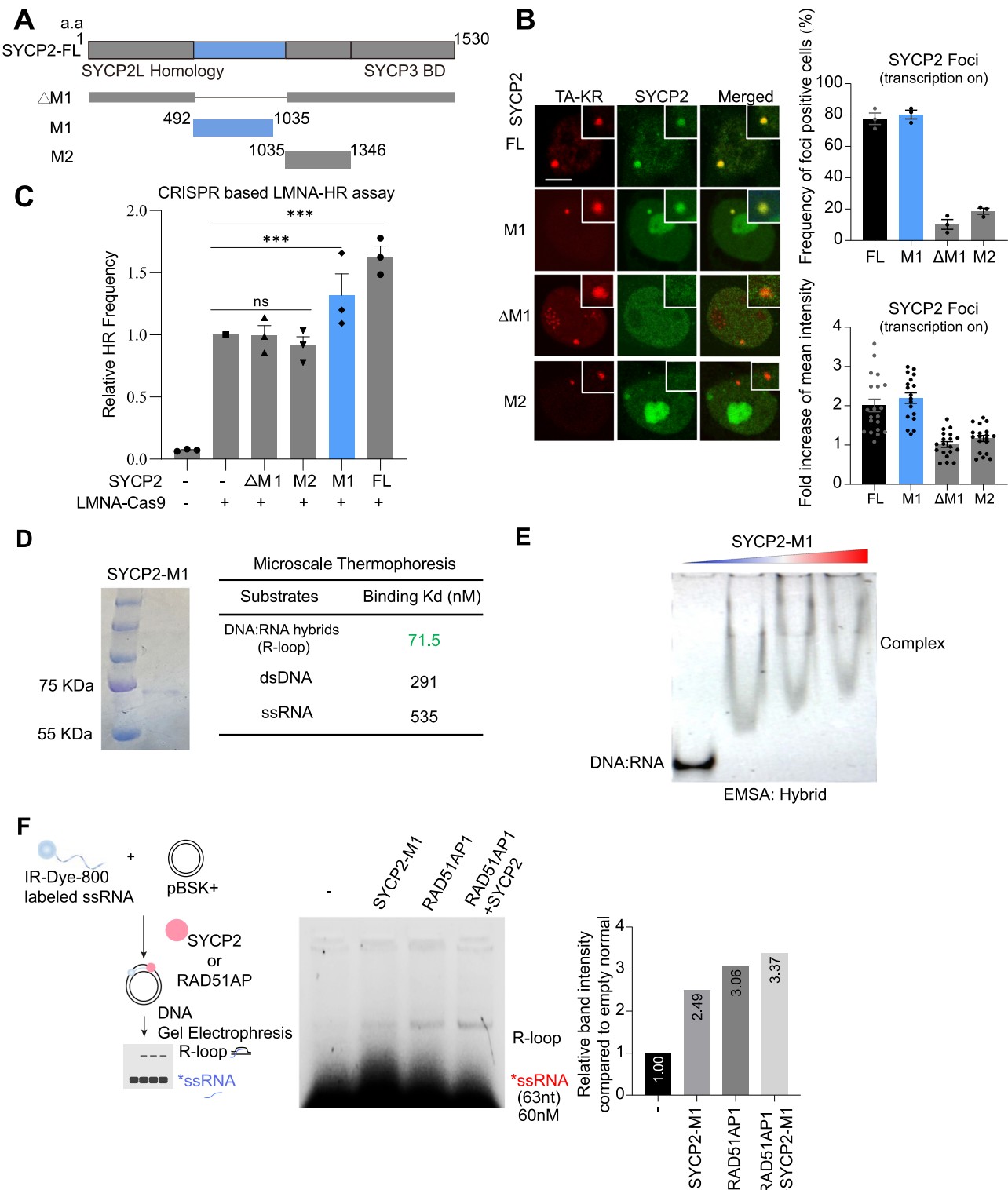

**Fig. 4 | SYCP2-M1 domain facilitates R-loop formation. A** Scheme of SYCP2 deletion and truncation constructs ΔM1, M1, and M2. **B** U2OS-TRE cells transfected with TA-KR and SYCP2-full length (FL), M1, ΔM1, and M2 were light-activated and recovered for 30 min. Frequency of SYCP2 foci positive cells at TA-KR ($n = 3$ experiments, Mean +/− SEM) and Fold increase of SYCP2 foci at sites of KR compared to background was quantified ($n = 20$ cells, Mean +/− SEM). The represented images of the protein recruitments at TA-KR foci were shown. **C** Relative HR frequency using CRISPR-based LMNA reporter assay in U2OS cells with or without indicated fragments of SYCP2 overexpression ($n = 3$ experiments, Mean +/− SEM). **D** 80 ng purified M1 protein was loaded for the SDS-PAGE analysis followed by Coomassie staining. Summary of the binding Kd value of 10 nM

labeled SYCP2-M1 protein with 0.5 μM nucleic acid substrates measured with Microscale thermophoresis (MST). **E** The binding of purified SYCP2-M1 protein (at 5.3, 8.9,12.4 μM) with 0.1 μM DNA-RNA hybrids was analyzed in EMSA. The experiments were repeated three times with similar results. **F** Scheme of In Vitro R-loop formation assay is shown on left. 2.6 μM labeled ssRNA and 30 nM pBSK+ plasmid was incubated with or without 0.15 μM Purified SYCP2-M1 protein or 0.15 μM Rad51AP1 for 20 min, and the formation of R-loops was analyzed with 1% TAE agarose gel. The relative intensity of the band intensity in each group compared to empty control was measured. Statistical analysis was done with the unpaired two-tailed Student-t-test, ****$p < 0.0001$. Scale Bar = 10 μm. Source data are provided as a Source Data file.

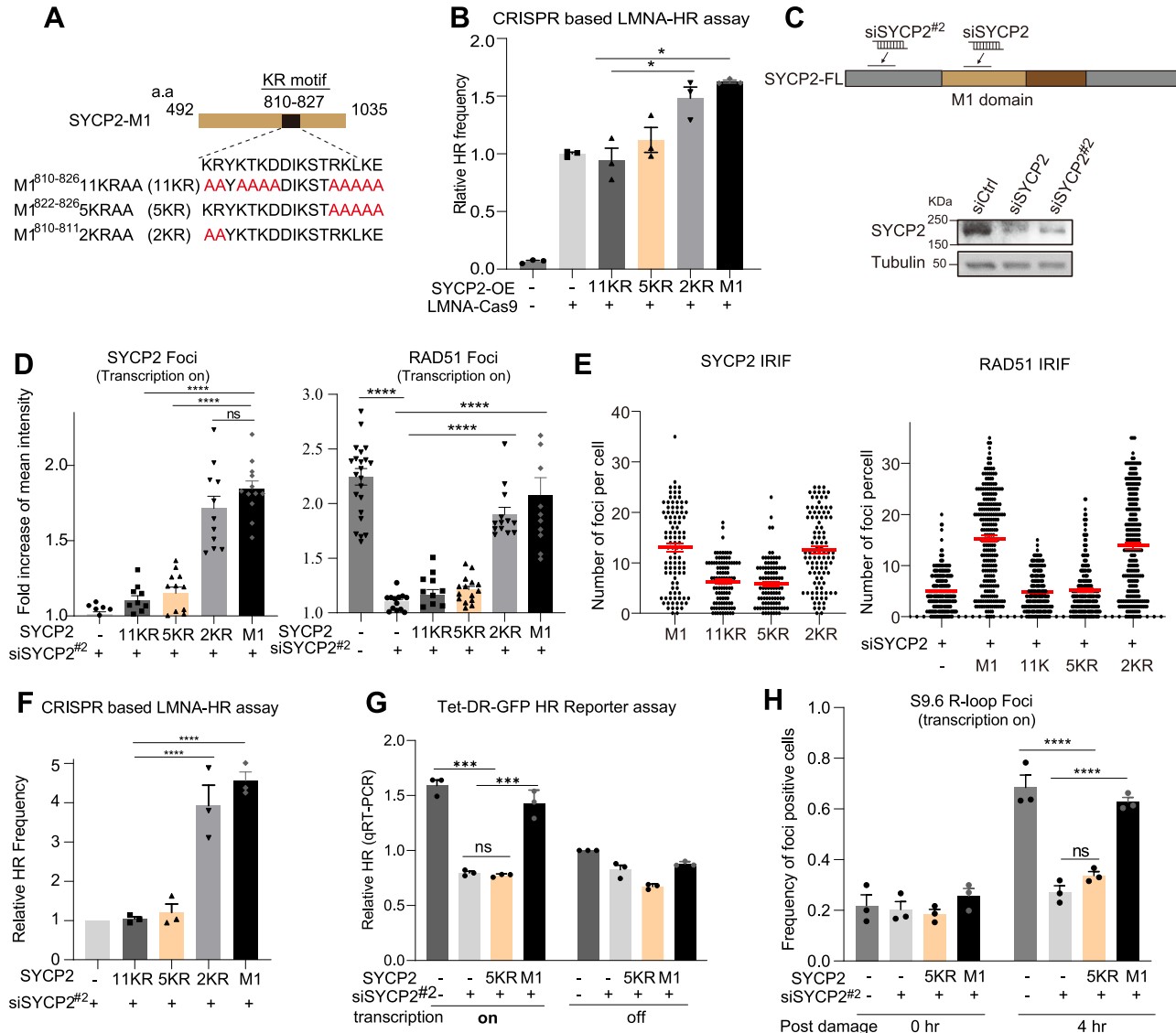

**Fig. 5 | Lysine(K)/Arginine(R) motif in the SYCP2 M1 domain is required for R-loop stabilization and TC-HR. A** Schematic of SYCP2 mutants by replacing Lysine(K)/Arginine(R) with Alanine(A) in the M1 domain of SYCP2. **B** Relative HR frequency using CRISPR-based LMNA reporter assay with overexpression of indicated SYCP2 and SYCP2 mutants ($n = 3$ experiments, Mean +/− SEM). ($p = 0.0177$). **C** Schematic and WB of siSYCP2 and siSYCP2#2 were shown. The experiments were repeated three times with similar results. **D** Fold increase of intensity of SYCP2 (Left) or RAD51 (right) at TA-KR sites in siSYCP2#2 pretreated U2OS cells with or without expression of SYCP2 or its mutant as indicated ($n = 6$ cells and $n = 11$ cells, Mean +/− SEM). **E–G** The numbers of SYCP2 and RAD51 IRIF ($n = 200$ cells, +/− SEM)

(**E**); relative HR frequency in CRISPR-based LMNA HR assay (**F**) ($n = 3$ experiments, Mean +/− SEM), and relative TC-HR frequency (**G**) ($n = 3$ experiments, Mean +/− SEM) in indicated cells were shown. Mean frequency of the HR (FACs) compared to the siSYCP2 is shown. ($p = 0.0001$, $p = 0.0002$). **H** siSYCP2#2 pretreated U2OS-TRE cells transfected with TA-KR were light-activated and recovered at 4 hr and stained with anti-S9.6. The frequency of positively stained S9.6 foci at TA-KR was quantified ($n = 3$ experiments, Mean +/− SEM). Mean frequency of quantity of the positive staining of S9.6 at TA-KR is shown. Statistical analysis was done with the unpaired two-tailed Student-t-test, ****$p < 0.0001$. Source data are provided as a Source Data file.

independently of each other (Supplementary Fig. 6B). In the TCGA database, SYCP2 expression is not significantly different in tumors with or without *BRCA1/2* mutations (Fig. 6C, Supplementary Fig. 6C). Taken together, these results suggest that SYCP2's expression, localization, and contribution to DDR resistance are not significantly influenced by the BRCA1/2 status.

To further investigate if SYCP2 can function independently of BRCA1, we tested HCC1937 and HCC1954, two breast cancer cell lines defective for BRCA1. The levels of IRIF of RAD51 in HCC1937 and HCC1954 cells were significantly lower than those in BRCA-proficient U2OS cells, but substantial levels of IRIF of RAD51 remained detectable in HCC1937 and HCC1954 cells (Fig. 6D, Supplementary Fig. 6D). SYCP2 KD in HCC1937 and HCC1954 cells further reduced RAD51 IRIF without affecting the expression of BRCA1/

2 (Fig. 6D). Reversely, SYCP2 OE in HCC1937 and HCC1954 cells increased RAD51 IRIF (Fig. 6E). SYCP2 KD also reduced RAD51 IRIF both in WT and siBRCA1 treated HeLa cells (Supplementary Fig. 6E), supporting the notion that SYCP2 is able to promote RAD51-mediated repair independently of BRCA1.

TOP1i, such as CPT and CPT11, are clinically used in the treatment of HR-proficient breast cancers[33]. Given that the expression levels of *SYCP2* strongly correlate with the IC50 of CPT11 (Fig. 1), we asked whether SYCP2 expression affects the CPT11 response independently of BRCA1. siBRCA1 or siSYCP2 alone sensitized cells to CPT11 (Fig. 6F). Combined siBRCA1 and siSYCP2 treatment further sensitized cells to CPT11 (Fig. 6F), suggesting that BRCA1 and SYCP2 have independent functions in promoting cellular resistance to TOP1i. Similarly, combined siBRCA1 and

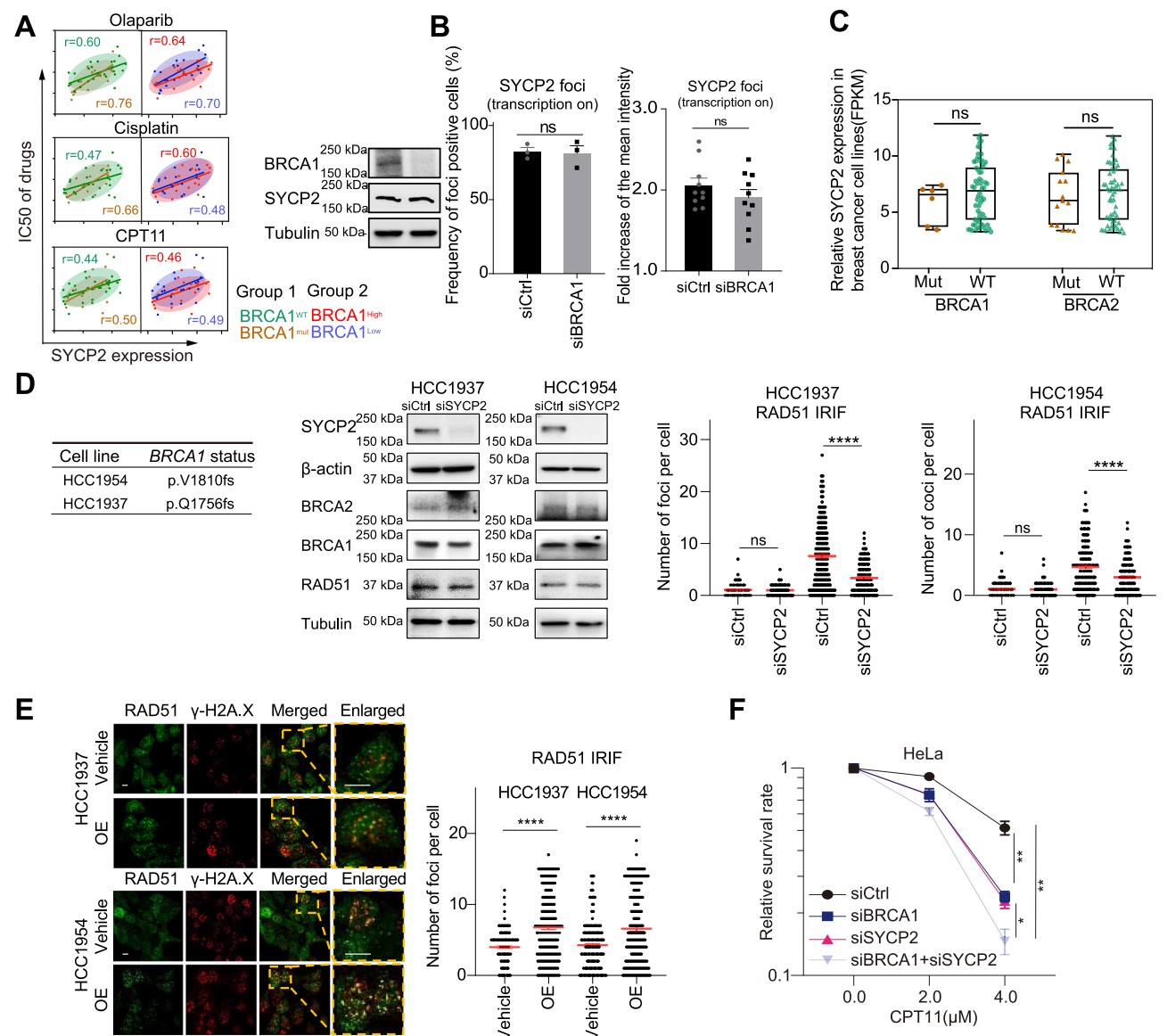

**Fig. 6 | SYCP2 promotes TC-HR independently of BRCA1. A** Plot of correlation between SYCP2 expression and IC50 of Olaparib, Cisplatin, and CPT11 in groups of BRCA WT vs. mutant cells (left), high BRCA1 vs. low BRCA1 (right). **B** siBRCA1 or siCtrl pretreated U2OSTRE cells transfected with GFP-SYCP2 and TA-KR were light-activated and recovered for 30 min. The frequency of foci-positive cells was quantified ($n = 3$ experiments, Mean +/− SEM). Fold increase of SYCP2 at TA-KR sites was quantified ($n = 10$ cells, Mean +/− SEM). **C** Comparison of SYCP2 RNA expression from CCLE database in BRCA1/2 mutant or proficient breast cancer cell lines ($n = 6$ samples, $n = 65$ samples, $n = 15$ samples, $n = 59$ samples). The analysis was normalized to Fragments Per Kilobase Million (FPKM). **D** IRIF of RAD51 1 hr after 2 Gy IR with or without siSYCP2 were quantified in HCC1954 and HCC1937 cells

($n = 200$ cells, Mean +/− SEM). Mean quantity of IRIF foci per cell is shown. WB of SYCP2, BRCA2, BRCA1 and RAD51 in HCC1937 and HCC1954 in siCtrl or siSYCP2 treated cells was shown. **E** IRIF of RAD51 1 hr after 2 Gy IR in HCC1954 and HCC1937 cells with empty vector or SYCP2 OE were quantified ($n = 200$ cells, Mean +/− SEM). Mean quantity of IRIF foci per cell is shown. **F** Cell survival rate of HeLa cells with siCtrl, siBRCA1, siSYCP2, or siBRCA1+siSYCP2 via colony-forming assay with the treatment of CPT11 at indicated dose ($n = 3$ experiments, Mean +/− SEM). ($p = 0.0026$, $p = 0.0383$, $p = 0.0011$) Statistical analysis was done with the unpaired two-tailed Student-t-test, ****$p < 0.0001$. Scale Bar = 10 μm. Source data are provided as a Source Data file.

siSYCP2 treatment sensitizes cells to Cisplatin or PARPi more efficiently compared to each siRNA alone (Supplementary Fig. 6F). Moreover, signals of S9.6 following the treatment of PARPi or TOP1i were suppressed by siSYCP2 (Supplementary Fig. 6G), indicating SYCP2-dependent R-loop formation contributes to repair. It should be noted that although our results suggest that SYCP2 has an ability to promote RAD51-mediated repair independently of BRCA1, they do not exclude the possibility that SYCP2 facilitates BRCA-mediated HR when BRCA1/2 are present. Nonetheless, the ability of SYCP2 to promote repair independently of BRCA1 may contribute to the resistance of BRCA-deficient tumors to DDR drugs.

## Relevance of SYCP2 in the DDR drug response in vivo

To test whether SYCP2 KD affects the DDR drug response in vivo, MDA-MB-231 cells infected with lentiviruses (LV) expressing siSYCP2 or siControl (siCtrl) were injected intraperitoneally into mice, and tumor growth was measured over 23 days (Fig. 7A). No significant alterations in body weight of the mouse were observed across the treated groups (Supplementary Fig. 7A). Importantly, PARPi reduced the growth of SYCP2 KD tumors more than control tumors (Fig. 7A), showing that SYCP2 depletion enhances the response of tumors to PARPi in vivo.

Given the level of ER varies in breast cancer patients, we analyzed the correlation between the status of ER levels with SYCP2 expression and the outcome of hormone treatment (Tamoxifen) with SYCP2

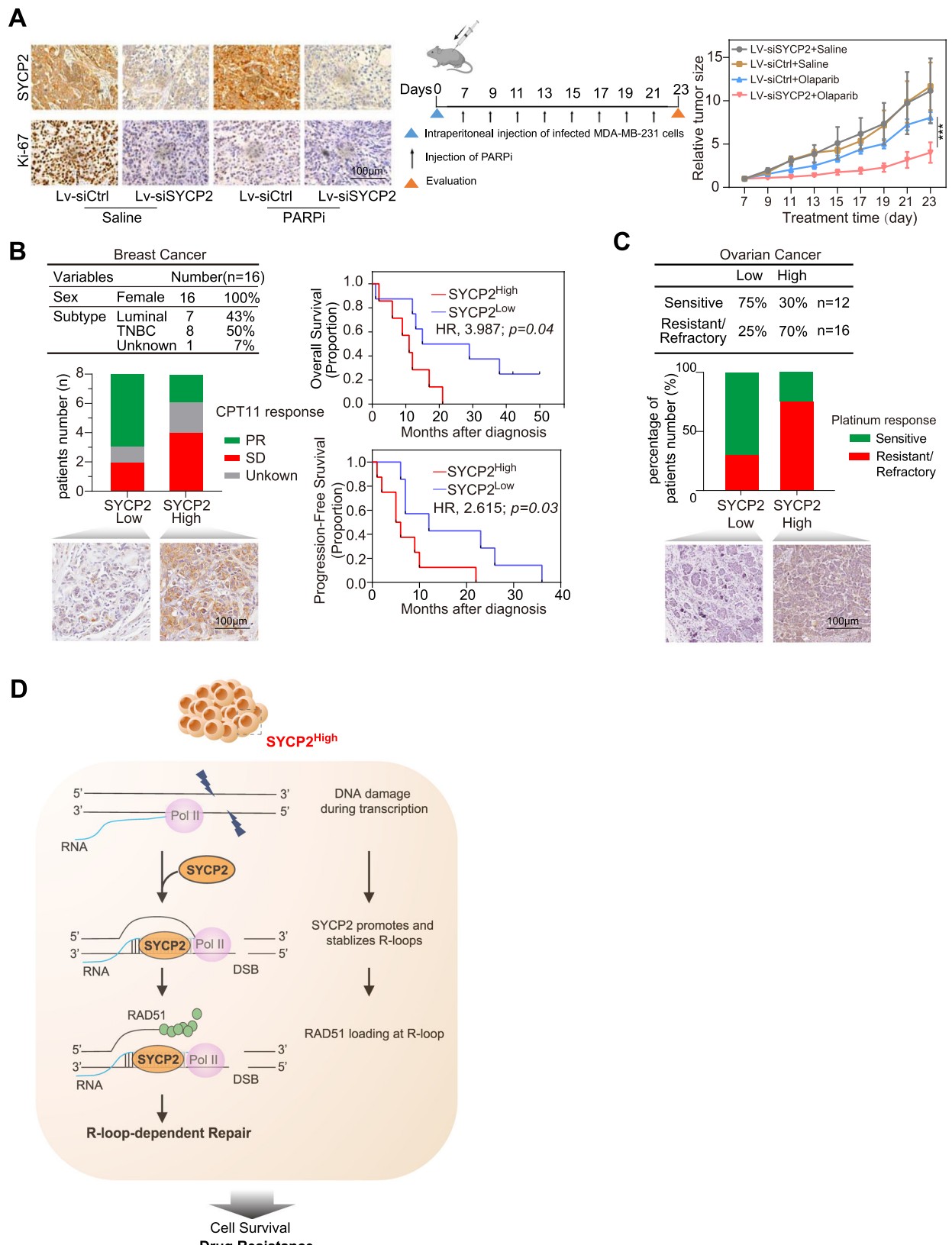

expression. There is no obvious correlation between ER status and SYCP2 expression levels (Supplementary Fig. 7B). Similarly, there is also no significant correlation between the outcomes of Tamoxifen treatment and SYCP2 expression (Supplementary Fig. 7C). Based on the *SYCP2* mRNA data from a large number of tumors, we found that the mRNA levels of *SYCP2* in breast tumors do not significantly correlate with tumor stages (T1-T4), pathological stages (Stage I-IV), lymph node status (N0-N3), and Her2 levels (Supplementary Fig. 7D). Furthermore, *SYCP2* mRNA levels are not significantly different in various subtypes of breast cancers (Supplementary Fig. 7E). The fact that *SYCP2* expression does not correlate with various markers of tumor progression suggests that SYCP2 upregulation is not simply a

**Fig. 7 | SYCP2 is a predictive biomarker for drug response in patients. A** MD-MBA-231 cells infected with lentiviruses (LV) expressing siSYCP2 or siCtrl were injected intraperitoneally into mice. Mice were given Saline and 50 mg/kg Olaparib. The representative images of IHC staining of SYCP2 of tumor tissues in the tested mice groups were shown (left). Tumor volume was measured over 23 days (right). (*p* = 0.0005). Created with BioRender.com. **B** Summarization of subtypes of breast cancer patients in a TOP1i [Sacituzumab Govitecan (IMMU-132)]-treated group. Patients were divided into the SYCP2[high] group and the SYCP2[low] group. The cutoff 12 value is the median. Representative images of IHC staining of SYCP2 were shown in each group. Numbers of patients' responses as Partial Response (PR) or Stable Disease (SD) in the SYCP2[high] group and SYCP2[low] group were shown, Kaplan-Meier curves of patients' overall survival and progression-free survival were shown on the right. The analysis was done in one-sided comparison. **C** Summarization of retrospective study in SYCP2 expression in ovarian cancer patients. Patients were divided into the SYCP2[high] group and the SYCP2[low] group based on the SYCP2 staining results. Representative images of IHC staining of SYCP2 were shown in each group. Numbers of patients' responses to Platinum Sensitive or Platinum-Resistant/Refractory tumor in the SYCP2[high] group and SYCP2[low] group were shown. **D** The scheme of the role of SYCP2 in contributing to HR and drug resistance in cancer. Source data are provided as a Source Data file.

consequence of tumor progression. Instead, DNA hypomethylation of the *SYCP2* gene during tumorigenesis is likely an event that enables tumor cells to tolerate genomic instability (Fig. 1E).

To test whether SYCP2 levels are relevant to the DDR drug response in patients, we analyzed 16 breast cancer patients (7 luminal ER/PR positive and 8 triple-negative with no known BRCA mutations, and 1 unknown BRCA status) treated with Sacituzumab Govitecan (IMMU-132), an antibody-conjugated TOP1i for breast cancer treatment[33]. We performed IHC analysis on tissue microarrays of tumors from these patients. We quantified the levels of SYCP2 in tumors with IHC and classified patients into SYCP2[high] and SYCP2[low] groups (Supplementary Fig. 7F). Tumors from the SYCP2[low] patients displayed SYCP2 staining in < 50% cells, whereas tumors from the SYCP2[high] patients displayed SYCP2 staining in >50% cells. Patients with SYCP2[high] tumors had a reduced rate of partial response (PR) but an increased rate of stable disease (SD) (Fig. 7B), suggesting that high SYCP2 expression in tumors is associated with resistance to TOP1i. High SYCP2 expression is also associated with decreased overall survival (OS) and reduced progression-free survival (PFS) among patients (Fig. 7B right). Furthermore, the clinical data of 1,095 breast cancer patients in the TCGA database confirms that high SYCP2 expression is associated with poor prognosis (Supplementary Fig. 7G, H). Thus, high SYCP2 expression in breast tumors is a potential marker for poor TOP1i response and poor prognosis in patients.

Finally, we analyzed 28 ovarian high-grade serous tumor samples. All these patients received platinum-based chemotherapy and the progression-free interval was determined. From IHC staining, we classified patients into SYCP2[low] and SYCP2[high] groups based on the amount of positive staining and intensity of SYCP2 using a similar method as in the analysis of breast cancer samples. A larger portion of the ovarian cancer patients in the SYCP2[high] group was resistant to platinum-based therapy compared to the SYCP2[low] group (Fig. 7C). This retrospective analysis suggests that SYCP2 could serve as a predictive biomarker for the platinum response in ovarian cancer.

## Discussion

In this study, we show that SYCP2 is commonly and aberrantly expressed in several cancer types, including breast, ovarian and several other cancers. Furthermore, the levels of SYCP2 expression in cancer cells are broadly associated with resistance to DDR drugs. In cancer cells expressing SYCP2, SYCP2 clearly contributes to DSB repair. We propose that, although SYCP2 is not normally expressed outside of meiosis, the aberrant expression of SYCP2 in cancer cells augments DSB repair and allows them to cope with genomic instability. Consistent with this idea, in cancer cells expressing SYCP2, a faction of the HR activity is dependent on SYCP2, showing that the HR pathway in these cancer cells has become partially SYCP2-dependent. This SYCP2-augemented HR pathway in cancer cells not only allows them to survival intrinsic DNA damage, but also makes them resistant to the DNA damage induced by DDR-targeted drugs, providing a potential therapeutic target and a biomarker for DDR therapy response in cancers.

Our results also provide insights into the mechanisms by which SYCP2 promotes HR. SYCP2 is recruited to DNA damage sites in a transcription-dependent manner, and it is required for the efficient localization of RAD51 to DSBs, suggesting that SYCP2 functions in the TC-HR pathway. Importantly, we find that SYCP2 is required for the efficient accumulation of R-loops at DNA damage sites in cells, and that SYCP2 is sufficient to promote R-loop formation in vitro, suggesting that SYCP2 augments TC-HR by facilitating R-loop accumulation (Fig. 7D). Several specific KR residues in the M1 regions of SYCP2 are required for binding and the localization of SYCP2 to DNA-RNA hybrids and R-loops, the efficient localization of RAD51 to DSBs, and the ability of SYCP2 to augment HR. Although persistent DNA-RNA hybrids inhibit canonical HR[34], accumulating evidence show that DNA-RNA hybrids and R-loop could trigger TC-HR[10,12,13,16], in which DNA-RNA hybrids recruit DNA repair proteins differently from canonical HR.

Our previous studies suggest that damage-induced R-loops stimulate RAD51 recruitment[10] and RAD51-mediated D-loop formation[12], which may explain the positive effects of SYCP2 on HR (Fig. 7D). In addition to its role in loading RAD51 to DSBs, we found that SYCP2 is also required for protecting nascent DNA from degradation at stalled replication forks (Supplementary Fig. 8A), which is known to be a RAD51-dependent process[35]. Recent studies by others and us suggested that the degradation of nascent DNA at stalled forks and single-stranded DNA (ssDNA) gaps generated at stalled forks may contribute to the PARPi sensitivity of BRCA-deficient cells[36–38]. It is possible that SYCP2 confers PARPi resistance by promoting RAD51 loading to stalled forks or ssDNA gaps, providing an attractive hypothesis to test in future studies.

Others and we previously showed that temporary transcription repression induced by damage is associated with efficient R-loop-dependent repair[10,12,13,16]. Moreover, a recent study has highlighted the role of BMI-1-dependent transcriptional inhibition in promoting DNA end resection and HR[39]. Although the function of SYCP2 is not associated with end resection at DSBs, its role in HR is transcription-related. We also assessed whether the level of SYCP2 impacted overall transcriptional levels. We observed that the mRNA levels were significantly lower in the SYCP2 knockdown group compared to the siCtrl group. Conversely, overexpression of SYCP2 increased the mRNA levels compared to vehicle control. These findings suggest that SYCP2 expression in cancer cells might promote DSB-transcriptional collisions by upregulating transcriptional events (Supplementary Fig. 8B). In addition, RNA polymerase inhibitors and RNaseH1 abolished SYCP2 mediated RAD51 IRIF, reconfirming the SYCP2-RAD51-mediated repair is transcription and R-loop-dependent. The importance of SYCP2 is also reflected by the level of chromosomal aberrations with or without SYCP2. We observed an increased rate of chromosomal aberrations in siSYCP2-treated cells and a slightly lower level of chromosomal aberrations in the SYCP2-overexpressing HeLa cells compared to the vehicle group (Supplementary Fig. 8C). These results support the notion that SYCP2 plays a role in transcription-related maintenance of chromosomal integrity.

This study reveals an intricate relationship between BRCA1/2 and SYCP2. BRCA1/2 are ubiquitously expressed in normal cells and required for the canonical HR pathway. In contrast, SYCP2 is aberrantly expressed in cancer cells and augments TC-HR. On one hand, in cancer cells expressing SYCP2, loss of SYCP2 results in a significant reduction in HR activity, suggesting that SYCP2 may facilitate BRCA1/2-mediated HR. On

the other hand, the association of SYCP2 with resistance to DDR drugs is not affected by the BRCA status in tumors, and SYCP2 promotes RAD51 focus formation even in BRCA1-deficient cells, suggesting that SYCP2 has an ability to function in TC-HR independently of BRCA1/2. It is known that lack of 53BP1 promotes HR[40,41], in addition to BRCA1, we also examined whether SYCP2 is required for HR in BRCA + 53BP1 double KD cells. We found that SYCP2 is required for HR in BRCA and 53BP1 double KD cells (Supplementary Fig. 8D left). Both in the presence and absence of 53BP1, overexpression of SYCP2 contributes to increased HR activity (Supplementary Fig. 8D right), suggesting that SYCP2 contributes to drug resistance independently of both BRCA1 and 53BP1. Thus, we speculate that SYCP2 can facilitate HR through both BRCA-dependent and -independent mechanisms by promoting R-loop accumulation. Both functions of SYCP2 may contribute to the resistance of cancer cells to DDR targeted drugs regardless of the BRCA status. According to this hypothesis, targeting SYCP2 in tumors expressing SYCP2 may overcome the resistance to DDR drugs, including PARPi and TOP1i, independently of the BRCAness.

Our findings that SYCP2 is commonly and aberrantly expressed in cancer cells and associated with resistance to DDR drugs draw attention to the dysregulation of meiotic protein in tumors. It is worth noting that SYCP2 is a meiotic protein involved in the pairing and recombination between homologous chromosomes[19,42]. Upregulation of meiotic DNA recombination proteins may be a common mechanism conferring DNA damage resistance in tumors. The rewiring of DNA repair/recombination pathways might serve as an important mechanism driving the resistance to DDR drugs, revealing a similarity between the rewiring of DNA repair pathways and oncogenic signaling pathways in cancer. For example, both RAD51 and the meiotic recombinase DMC1 function in meiosis[43], and the upregulation of RAD51 function and DMC1 has been linked to PARPi resistance[44]. It is conceivable that SYCP2 promotes RAD51 function when SYCP2 is aberrantly expressed in mitotic cells. Moreover, cohesion proteins, such as STAG2 and Rad21, are important for meiosis and also bind R-loops[45,46]. Although DNA-RNA hybrids inhibit canonical HR[47], accumulating evidence show that DNA-RNA hybrids and R-loop could trigger TC-HR, in which DNA-RNA hybrids recruit DNA repair proteins differently from canonical HR. Here, we observed that SYCP2 binds to DNA-RNA hybrids and promotes R-loop formation. These observations reveal a previously unknown function of meiotic proteins at sites of R-loops. It would be important to understand the network of aberrantly expressed meiotic proteins in cancer cells in future studies. In normal mitotic cells, such a network might not be essential for cell survival due to the repression of meiotic genes. However, in cancer cells suffering oncogenic stress and increased genome instability, the abnormal expression of SYCP2 and perhaps other meiotic proteins may augment TC-HR and allow cancer cell survival. In particular, SYCP2 expression in cancer might enable cells to tolerate genomic instability, promoting tumorigenesis at the early stage and priming cells to DDR drug resistance during tumor progression.

Recent studies have revealed several factors that regulate the TC-HR pathway. For example, at nucleases-generated DSBs, RAD51AP1 facilitates R-loop formation[12]; At sites of ROS-induced DNA damage, which includes both DNA single- and double-strand breaks, RPA1, CSB, and RAD52 bind to R-loops and facilitate RAD51-mediated DSB repair[10,13,48]. In future studies, it is important to investigate how SYCP2, as an R-loop-forming and -binding factor, functions with other TC-HR proteins to repair DSBs. The specific expression of SYCP2 in tumors and its contribution to DDR drug resistance makes the TC-HR pathway an attractive biomarker for cancer diagnosis, a predictor of the DDR drug response, and a potential target for therapy.

## Methods
### Database
Public RNA sequencing (RNAseq) data for cell lines were obtained from the CCLE (Cancer Cell Line Encyclopedia) project (https://portals.broadinstitute.org/ccle) and GDSC (Genomics of Drug Sensitivity in Cancer) project (https://www.cancerrxgene.org/). All expression data were processed to TPM by using Python. Expression data from CCLE was normalized by using ln (TPM + 1). RNAseq data of the gene expression are available for download at TCGA (https://www.cancer.gov/), GTeX (https://gtexportal.org/), TARGET (https://software.broadinstitute.org/cancer/cga/target), and treehouse (https://treehousegenomics.soe.ucsc.edu/public-data/). The full names of all types of cancer used in the analysis are shown in Supplementary Table 1.

### Cell culture, plasmids, siRNAs, and chemicals
HCC1954, HCC1937, MDA-MB-231, and HeLa cells were from ATCC and cultured in Dulbecco's modified Eagle medium (DMEM, Lonza, Catalog#12-604 F) and Roswell Park Memorial Institute (RPMI) 1640 Medium (Sigma, R8758) with 10% (vol/vol) fetal bovine serum (FBS, XY Cell Culture, FBS-500) at 37 °C, 5% CO2. ZR75 cells were cultured in Dulbecco's modified Eagle medium (DMEM, Lonza, Catalog#12-604 F) with 15% (vol/vol) FBS (XY Cell Culture, FBS-500) at 37 °C, 5% $CO_2$. The U2OS-TRE cell line was derived from wild-type U2OS cells (ATCC) by inserting an array of TRE/I-SceI and a transcription cassette in the genome. Plasmids pBROAD3/TA-KR, tetR-KR, TA-Cherry, tetR-Cherry, and pEGFP-RAD52 were used in our previous study[10]. pEGFP-SYCP2, CMV-SYCP2-myc-DDK, pEGFP-C3 SYCP2 fragments 492–1035, 1036–1346 were cloned into pEGFP-C3 vectors with KpnI and BamHI. The 510–960 fragment has an added NLS sequence in the N terminus to ensure nucleus localization. The 2KR, 5KR, and 11KR mutants in the SYCP2 492-1035 (SYCP2-M1) fragments were created using DNA synthesis from Geneuniversal. Plasmids were transfected by Lipofectamine2000 (Invitrogen, 11668019) using a standard protocol. siRNAs were transfected with Lipofectamine RNAiMax (Invitrogen, 13778150) 48–72 h before analysis. The siRNAs used in this study are siSYCP2 (Integrated DNA technologies: GUCCAAGGAAUCAUGAUGAACUUAA), siSYCP2#2 (TGGCATGCTTGGAGACAAA), siBRCA1 (L-003461-00, Dharmacon), and siBRCA2 (GS675, Qiagen), siRPA (H00140885-R01, abnova), siMRE11 (Dharmacon: GAGCAUAACUCCAUAAGUAUU & CCUGGUUGUUGUAGUAAGAUU), siCtIP (Dharmacon: TCCACAACATAATCCTAATAA & GCUAAAACAGGAACGAAUC & AAGCUAAAACAGGAACGAAUC), si53BP1 (AGAACGAGGAGACGGUAAUAGUGGG & GAGAGCAGATGATCCTTTA). Cisplatin (Sigma, 1134357), PARPi Olaparib (AZD2281/Ku-0059436, Sellekchem, S1060), Irinotecan hydrochloride (CPT11, Sigma, I1406) were used at the indicated dose.

### Colony-forming assay
Approximately 400 cells were replated on 6 cm dishes 24–48 h after siRNA transfection. Cells were incubated in DMEM (10% FBS) containing olaparib or cisplatin for 9 days. For treatment with ionizing radiation (IR), cells were washed and irradiated with the indicated dose 8 h after passaging. Colonies were stained with 0.3% crystal violet/methanol and counted 8–10 days after treatments. Each experiment was performed 3 times and the standard derivation (SD) was calculated and indicated in the graphs. Results were normalized for plating efficiencies.

### Laser micro-irradiation
The Olympus FV1000 confocal microscopy system (Cat. F10PRDMYR-1, Olympus) and FV1000 software were used for the acquisition of images. Cells were cultured in 35 mm glass-bottom dishes (MatTek, P35GC-1.5-14-C) before observation. The damage was induced with a 405 nm laser. The laser passed through a PLAON 60X oil lens. Cells transfected with GFP-tagged proteins were incubated at 37 °C on a thermos plate in normal media during observation. For the evaluation of accumulation and kinetics, the mean intensity of each accumulated point or line was obtained after subtraction and quantified by ImageJ.

## DART assay

DART assay has been used and described in previous studies[24]. In the DART assay, an array of TRE and a transcription cassette were integrated at one chromosomal locus of U2OS-TRE cells. The KillerRed (KR) protein is a fluorophore, which releases ROS upon 550–590 nm light exposure. KR or mCherry (non-damage control) were fused to tet-Repressor (tetR) or tetR with transcription activator (TA). TA-KR/tetR-KR/TA-mCherry/tetR-mCherry were transfected into U2OS-TRE cells. KR was activated in bulky cells by exposing cells to a 15 W Sylvania cool white fluorescent bulb for 25 min in a stage UVP (Upland, CA) for damage induction. Cells were then recovered for 30 min–1 h before live-cell observation or fixation. For γH2AX staining, cells were recovered for 12, 24, and 48 h before fixation. For I-SCEI induced damage, cells were co-transfected with pCMV-I-SceI plasmid and TA-Cherry, and incubated for 36 h before harvest[10]. The mean intensity was calculated by dividing the measured intensity of the selected area that colocalized with KR foci by ImageJ 1.52i software over the same-size arbitrary selected three areas in the nucleus.

## tetR-DR-GFP-Assay for measuring TC-HR

DR-GFP, EJ5, EJ2, and SSA were performed as previously described in ref. 24 and in supplementary methods. In the tetR-DR-GFP-Assay[12], U2OS-tet-DR-GFP reporter cells were seeded in a 6-cm dish 24 h before siRNA transfection. Cells were either transfected with siRNA or siControl (siCtrl) over 6 h before plasmid transfection. 5 μg of I-Sce-T2A-mCherry Plasmid was transfected using Lipofectamine 2000 according to the manufacturer's instructions. After 8 h incubation, cells were then trypsinized and plated into two 6 cm dishes. One plate was induced by adding 1 μg/ml of Doxycycline and another with a vehicle for control. After 72 h incubation, cells were collected. Half of the sample was used for flow cytometry analysis, while the rest of the sample was isolated for RNA purification using the Purelink RNA mini-Kit (Invitrogen) and reverse transcript to cDNA using Quantinova Reverse Transcription kit (Qiagen). The concentration of the genomic DNA was measured by NanoDrop and diluted for real-time qPCR with the PowerUp SYBR Green Master Mix (Invitrogen) and performed using StepOnePlus™ Real-Time PCR System (Applied Biosystems). cDNA was amplified using primers (GGGCGATGCCACCTACG) and (GGTGTTCTGCTGGTAGTGGTCG) targeting repaired sceGFP and primers (CAGCAAGTGGGAAGGTGTAATCC) and (CCCATTCTATCATCAACGGGTACAA) targeting reference genomic locus. Reactions were triplicated in three biologically independent experiments. Each experiment was repeated three times.

## CRISPR-based LaminA (LMNA)-HR reporter assay

CRISPR-based LMNA-HR reporter assay is modified from CRISPR-based LMNA mClover assay[29]. In the assay, LMNA-sgRNA and CRISPR/Cas9 were used to create a DSB 28 nucleotides upstream to the translational start site of LMNA. pCBASce-mcherry-LMNA plasmid served as the template which is homologous to the sequence upstream and downstream to the break site generated by CRSPR/Cas9. U2OS cells were seeded into a 35 mm glass-bottom dish (MatTek, P35GC-1.5-14-C) and transfected with 0.6 μg of the pCBASce-mcherry-LMNA, 0.3 μg of the LMNA-sgRNA, and 0.5 μg GFP-tagged SYCP2 fragments by using FuGENE 6 transfection reagent (Promega, E2691). Puromycin (1:1000) was added after 24 h of transfection to select sgRNA expressed cells. Cells were fixed with 4% paraformaldehyde 48 h after selection. Images of randomly selected region (containing more than 20 cells in one vision) were acquired by the Olympus FV1000 confocal microscopy system. The frequency of mCherry-positive cells, which indicates successful repair at endogenous LMNA gene, was counted in GFP-SYCP2 expressed cells.

## Microscale thermophoresis (MST) assay

Purification of SYCP2-M1 is in supplementary methods. The purified M1 protein was labeled by the Monolith His-Tag Labeling Kit (Cat# MO-L018) according to the manufacturer's instructions. Briefly, 90 μL of protein (200 nM) and the same volume of His labeling dye (100 nM) were mixed together and incubated for 30 min at room temperature. The labeled protein was centrifuged at 4 °C for 10 min and transferred the supernatant to a fresh tube for the binding assay. The MST assay was performed on the Monolith NT.115 instrument in the Center for Macromolecular Interactions, Harvard Medical School. We used the PBST buffer provided in the kit, with a further addition of 10% Glycerol to conduct our assay. For the M1 and nucleic acids binding assay, the protein concentration was 10 nM, and the substrate concentration was 0.1–0.5 μM.

## Electrophoretic mobility shift assay (EMSA)

The 5′-End maleimide-IR800-labeled hybrid substrate was incubated with M1 in Buffer B (25 mM Tris-HCl, pH 7.5, 1 mM MgCl2, 1 mM DTT, 50 μg/mL BSA) with 50 mM NaCl for 15 min at 37 °C. Reactions were loaded on 6% PAGE-TBE gel and resolved at 4 °C. Gels were imaged using ChemiDoc Imaging System (Bio-Rad).

## R-loop formation assay

We performed this In Vitro assay as described by Ouyang J. et al.[12]. Rad51AP1 protein and IRDye-800 5′end labeled ssRNA, pBSK+ plasmid was used in the previous study[12]. Briefly, the labeled ssRNA was mixed with 0.15 μM SYCP2 M1 or 0.15 μM SYCP2 M1 and Rad51AP1 in freshly made buffer D (35 mM Tris-HCl pH 7.5, 1 mM DTT, 2 mM MgCl$_2$, 2 mM CaCl2, 2 mM ATP or AMP-PNP, 50 μg/ml BSA and 50 mM KCl). pBSK+ plasmid (30 nM) was then added to the reaction, and samples were incubated for 20 min (min) followed by the supplement of 1 mg/ml proteinase K, 0.5% SDS, and 0.5 mM EDTA with further incubation for 5 min to stop the reaction. Reaction products were resolved on 1% agarose gels with TAE buffer. Gels were imaged using ChemiDoc Imaging System (Bio-Rad).

## Immunofluorescence staining, R-loop staining, and western blots (WB)

Cells were seeded in a 35 mm glass-bottom dish (MatTek, P35GC-1.5-14-C). After the transfection and treatment of indicated dose of irradiation through a Precision X-Ray machine (PXi, X-RAD 225 Lite), cells were first rinsed with phosphate-buffered saline (PBS, BE17-516F) and fixed in 4% paraformaldehyde (PFA; Affymetrix, 19943 1 LT) for 15 min at room temperature. Then they were washed three times with PBS, and followed by permeabilization using 0.2% Triton X-100 in PBS for 15 min, and then washed three times with PBS. The cells were blocked by 5% bovine serum albumin (BSA) (SIGMA, A-7030) in PBS for 1 h at room temperature. Primary antibodies were diluted in blocking buffer and added for overnight incubation at 4 °C. After overnight incubation, the cells were washed three times with 0.05% PBST and incubated with secondary antibodies for 1 h at room temperature, including Alexa Fluor 405/488/594 goat anti-mouse/rabbit IgG conjugate (Abcam, 1: 3000). Finally, they were washed three times by 0.05% PBST and stained with DAPI (4′,6-diamidino-2-phenylindole; 1:1000 in PBS) for 5 min at room temperature. The primary antibodies for immunoassays were SYCP2 (PA5-66486, Invitrogen, 1:500) RAD51 (ab63801, Abcam, 1:100), γH2AX (JBW301, 05–636, EMD Millipore, 1:400), BRCA1 (D-9, sc-6954, Santa Cruz Biotechnology, 1:100), Cyclin A (sc27162, B8, Santa Cruz Biotechnology, 1:50), CtIP (#61142, clone14-1, Active Motif, 1:200), RPA(#2267, RPA70/1, Cell Signaling Technology, 1:200), pRPA (A300-246A, Bethyl, 1:100).

For S9.6 staining, the fixed and permeabilized cells were steaming in TE buffer (10 mM Tris-HCl, 2 mM EDTA, Ph=9.0) on the 95 °C heating blocks for 20 minutes for the purpose of antigen exposure. Then the cells were blocked by 5% BSA for 1 h at room temperature. Primary antibody α-9.6 (ENH001, Karafast) were diluted at 1:200 in 5% BSA and applied to the cells and incubated in 4 °C overnight. Alexa Fluor 488 goat anti-mouse were used for detection. Frequency of foci positive cells were counted in three groups of 30 individual cells per group.

For WB analysis, samples were boiled at 95 °C for 5–8 min in an SDS loading buffer. Then the samples were subjected to electrophoresis in 8–12% SDS-polyacrylamide gels and transferred to the polyvinylidene difluoride membrane. The membranes were blocked with 5% non-fat milk in PBS for 1 h before being incubated with the primary antibody at 4 °C overnight. The primary antibodies for WB used in this study are GFP (11814460001, Roche, 1:2000), SYCP2 (LS-C386874, LifeSpan BioSciences, 1:1000), RAD51 (ab63801, Abcam, 1:1000), BRCA1 (D-9, sc-6954, Santa Cruz Biotechnology, 1:100), BRCA2 (ab9143, Abcam, 1:1000) and β-actin (#3700, 8H10D10, Cell Signaling Technology, 1:1000). Then the cells were washed three to four times with 0.1% PBST and incubated with horseradish peroxidase (HRP)-conjugated secondary antibody (1:10,000) for 1 h at room temperature. The membranes were washed in 0.1% PBST four times before exposure. Chemiluminescent HRP substrate was purchased from Abcam (Catalog#: WBKLS0500). Images were acquired in a BIO-RAD Universal Hood II machine with ImageLab software.

### Xenograft study

Lentiviral (LV-SYCP2-RNAi or LV-NC-RNAi) transfected MD-MBA-231 ($6.0 \times 10^5$) were injected intraperitoneally into the BALB/c nude mice. The injected mice were then randomly divided into four groups ($n = 6$/group). After the injection of lentiviral transfected cancer cells, mice would develop a palpable tumor within a week. On day seven, 50 mg/kg Olaparib and/or saline was intraperitoneally injected into the xenograft tumors once every 2 days after day 7 for 8 treatments around 20 days. Mice were sacrificed on day 23. Tumors were harvested, then fixed, and embedded in paraffin. The embedded tumor was then sectioned into 4 μm slices. All animal experiments were approved by and conducted according to the guidelines established by the Institutional Animal Care and Use Committee at Massachusetts General Hospital with the protocol number 2003N000186. The sectioned slices were deparaffinized and rehydrated before staining. The rehydrated sections were then blocked and incubated with primary antibody SYCP2 (PA5-67554, Invitrogen) and Ki-67 (sc-23900, Santa Cruz Biotechnology), then detected using the Dako Envision two-step method of immunohistochemistry (Carpinteria, CA, USA).

### Immunohistochemistry (IHC) of patients' tissues

A tissue microarray comprising 16 breast cancer patients' fixed sample and tumor specimens were obtained from the Massachusetts General Hospital Cancer Center. The collected tumor specimens and adjacent normal tissue samples were fixed in 4% paraformaldehyde and stored in PBS. After sucrose infiltration, samples were filled with OTC and readied for cryosection. Ovarian tumor samples were obtained from patients who provided informed consent to an Institutional Review Board-approved banking trial (#07-049). The coded tumor samples along with their deidentified clinical data were obtained under a secondary use Institutional Review Board-approved protocol (#2014P002048). All samples were collected prior to treatment with chemotherapy and confirmed to be of high-grade serous histology by our institutional pathologists. All patients received platinum-based chemotherapy and the progression free interval was reported. Of the patients' tissue samples provided, 12 were platinum-sensitive and 16 were platinum-resistant or refractory. As described for the breast tissue, the samples were subjected to immunohistochemical staining for SYCP2, scored, and the percent positivity quantified as described above. The percent positive staining was correlated with either their platinum-sensitive or platinum-resistant/refractory status. The immunohistochemical staining, scoring, and analysis followed a triple-blinded manner. The group that prepared patients' samples labeled the samples with random numbers and was not involved in analysis. The pathology facility that performed IHC staining was not informed of patients' information and was not involved in quantification. Pathologists who quantified the expression did not perform IHC and were not informed of patients'

information. The images were captured at 20x and 40x through a digital slide scanner (Aperio CS2, Leica). The counting of positive cells and analysis was performed by one pathologist and one investigator separately in a blinded fashion. Cold sectioning and staining with Ki67 and SYCP2 antibodies (PA5-67554, 1:20) were performed by the Specialized Histopathology Services-MGH at Massachusetts General Hospital-East. Briefly, samples were fixed, embedded in paraffin, and sectioned into 4 μm thickness in animal study. After deparaffinization and rehydration, sections were blocked and incubated with antibodies against SYCP2 (PA5-67554, 1:20), Ki-67 (sc-23900, Santa Cruz,1:200), and then detected using the Dako Envision two-step method of immunohistochemistry (Carpinteria, CA, USA). All IHC staining was scored independently by 2 pathologists. We divided the positive staining results into 0–4 categories as following: 0: < 5%; 1: 6–25%; 2: 26–50%; 3: 51–75%; and 4: > 76% staining. SYCP2 low group contains categories 0 and 1, and the SYCP2 high group contains categories 3 and 4.

### Statistical analysis

The data were presented as mean ± SD from technical triplicates. Comparisons between each group were calculated using the Student-t-test, two-tailed Fisher's exact test method of summing small P-values, one-way and two-way analysis of variance, and Bonferroni's multiple comparison test as appropriate. A value of $P < 0.05$ was considered significant. GraphPad Prism version 7 was used for graphics (GraphPad Software, San Diego, CA, USA).

### Reporting summary

Further information on research design is available in the Nature Portfolio Reporting Summary linked to this article.

## Data availability

All datasets generated during and/or analyzed in this study are provided in the main manuscript and/ or its Supplementary Information files. Source data are provided with this paper.

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

## Acknowledgements

National Institutes of Health grant R01 GM118833 and CA282939 (LL). National Institutes of Health grant R01 CA181368, CA183976, and 1R21CA237964 (Xi. W). National Institutes of Health grant R01 GM076388 and CA197779 (L.Zo). National Institutes of Health grant P50 CA274158 (JO). Nile Albright Research Foundation and Vincent Memorial Hospital Foundation (B.R.R.).

## Author contributions

Conceptualization: LL; Methodology: YW, BG, LZh, XuW, XiZ, HY, FZ, XuZ, BZ, SY, AN, SL, JO, SBK, ELE, DZ, KL; Investigation: YW, BG, LuZ, XuW, XiZ, HY, FZ, XuZ, BZ, SY, AN, SL, JO, SBK, ELE, DZ, KL; Funding acquisition: LL, XiwW, BRR, L.Zo; Project administration: LL; Supervision: LL, BRR, LZo, YO, LWE, XS and XiW; Writing – original draft: YW, BG, and LL. All authors described their specific contributions and reviewed and edited the manuscript.

## Competing interests

The authors declare no competing interests.
