## [Peer Review File · Nature Communications]

Meiotic Protein SYCP2 Confers Resistance to DNA-Damaging Agents through R-Loop-Mediated DNA RepairREVIEWER COMMENTS

Reviewer #1 (Remarks to the Author):

In this manuscript, Wang Y. et al show that the meiotic protein SYCP2 is aberrantly expressed outside meiosis and upregulated in several cancer types, including breast and ovarian cancers known to be altered in homologous recombination (HR). In their analyses, SYCP2 upregulation in cancer cells results to be associated to resistance to treatments with DNA damaging agents, such as PARP1 and TOP1 inhibitors because SYCP2 stimulates DSB repair by HR. In addition, the authors try to demonstrate that SYCP2 functions at DNA breaks occurring at actively transcribed genes and sustains HR by promoting DNA:RNA hybrids formation. SYCP2 apparently stimulates RAD51 recruitment, while acting independently from BRCA1/2 status, enforcing HR at damaged transcribed loci.

The authors also identify the active motive of SYCP2 which is required to sustain HR and provide both in vitro and in vivo evidence that demonstrates how SYCP2 inactivation can strongly sensitize cancer cells to DNA damaging treatments. In addition, they propose SYCP2 abundance as a good biomarker for therapy targeting DDR in cancers.

I think the study is rich of data both in vitro and in vivo, is well written and clear, and definitely relevant for cancer treatment. Experiments are generally well designed, and conclusion are mostly agreeable. Therefore, I moderately support publication in Nature Communication after a step of revision. That is because, at this stage, I believe two important pieces of information are missing:

- 1) what recruits SYCP2 to site of DNA damage in cancer cells (are DNA damage sensors or transcription itself? Chromatin modification? H3K36 was previously suggested to stimulate HR for transcribed genes by Legube laboratory...);
- 2) In addition, it is not clear which is the mechanism by which SYCP2 might promote DNA:RNA hybrids formation and HR. Could it be that SYCP2 controls HR and hybrid formation by supporting resection?

I think the author should go deeper into these two aspects.

Please find below major and minor specific points

1) Authors show that SYCP2 KD reduces HR by DR-GFP reporter system while SYCP2 overexpression enhances it. Would be interesting to test also the impact of SYCP2 KD or its overexpression on NHEJ and alternative NHEJ with EJ5-GFP and EJ2-GFP respectively? Are other repair pathways affected as well?

2) Figure 2E: how does the undamaged cells look-like? Please show the undamaged cells for both siCTRL/siSYCP2. Are the indicated cells in S/G2? Please use a co-staining with Cyclin A and RAD51 to confirm it both in Figures 2 and Supplementary Figure 2C and D.

3) What about CTIP/BRCA1/pRPA recruitment after KD of SYCP2? Is the effect on RAD51 foci due to less resection? They should test efficacy of resection, for example by BrdU native staining after IR.

4) In the manuscript, data demonstrate that transcription inhibition reduces HR. Nevertheless, a recent paper suggests the opposite, showing that inhibition of transcription increases resection and HR ([https://www.cell.com/cell-reports/pdfExtended/S2211-1247\(22\)00277-7](https://www.cell.com/cell-reports/pdfExtended/S2211-1247(22)00277-7)). This paper should be mentioned and commented to highlight incongruencies.

5) The authors show that SYCP2 is recruited to ROS induced DNA damage in a transcription dependent manner. Is it occurring also at DSB induced by AsiSI restriction enzyme in the DivA cellular system that cut at active transcribed genes?

6) Since SYCP2 is known to be recruited at the level of transcribed loci before damage, its recruitment is not mediated by RNA or canonical RNA polymerase II machinery. What is the mechanism of SYCP2 recruitment to damage site then? Is MRN required?

7) Figure 3D: SYCP2 is required for DNA:RNA hybrids formation at damaged sites as detected by S9.6 DRIP. DNA end resection, is functional to induce DNA:RNA hybrids formation upon damage (see <https://pubmed.ncbi.nlm.nih.gov/30560944/>). Does resection take place to allow hybrid establishment in the presence of SYCP2 ? Which is the level of DNA:RNA hybrids in transcription OFF conditions as detected by S9.6 antibody both in the presence or in the absence of SYCP2? In this study: <https://elifesciences.org/articles/69881>, it was instead shown that hybrids inhibits HR. I think it should be discussed.

8) Figure 4B a single image might not be sufficient to state which SYCP2 fragment is required for efficient recruitment. A simple quantification with the percentage of cell might be added.

9) In Figure 5 D and E it should be shown also the transcription OFF condition, at least for the control sample with endogenous SYCP2.

10) Figure 6: it shows that SYCP2 acts independently from BRCA1/2 status. Is SYCP2 overexpressed in the cancer cell lines tested (HCC1937 and HCC1954)? Maybe the lack of activity of BRCA1/2 induces SYCP2 hyper activation to compensate lack of HR. It would be also important to check by western blotting if SYCP2 KD alters or not BRCA1/2 levels.

Minor points

Line 294 a reference is missing "(ref)"?

Reviewer #2 (Remarks to the Author):

The authors found an abnormally high expression of SYCP2, which is a component of the synaptonemal complex during meiosis, in breast and ovarian cancer cells. SYCP2 expression levels are associated with the drug resistance of several different PARP inhibitors, Cisplatin, and Topoisomerase I inhibitors. The authors showed that SYCP2 overexpression enhances the frequency of homologous recombination (HR), and SYCP2 depletion in cancer cells overexpressing SYCP2 impairs RAD51 foci formation and HR. Based on several biological experiments, the authors proposed a model in which SYCP2 overexpression in cancer cells confers resistance to chemotherapeutic agents by stimulating R-loop-mediated HR repair because HR generally contributes to cell survival through accurate DSB repair. The finding regarding the correlation between SYCP2 and drug resistance is important, and the proposed model is interesting; however, there are multiple concerns regarding the analysis and interpretation of this study.

1. One of the critical points that lead to confusion is the mechanism by which SYCP2 expression enhances HR in mitotic cells. The authors proposed a model in which SYCP2 overexpression enhances the generation of the R-loop at the DSB site in a transcription-dependent manner. If the formation of the R-loop is dependent on ongoing transcription, the increase in R-loop of SYCP2 overexpression cells may be caused by the upregulation of transcription events. Does SYCP2 overexpression upregulate the number of transcription events per cell, which increases the opportunity for DSB and transcription collision? Has the overall transcriptional activity been examined in SYCP2-expressing cells?

2. Another critical question is whether enhanced HR maintains the fidelity of DSB repair. For example, RAP80 depletion enhances DSBs undergoing resection but confers error-prone repair. The authors should examine genome stability following SYCP2-dependent HR to clarify whether increased cell viability (drug resistance) is associated with less genome instability, e.g., by chromosomal aberration assay.

3. In the model, the authors proposed that SYCP2 promotes R-loop formation at TC-DSB sites without enhancing resection. However, the authors did not examine RPA IRIF. The authors should confirm that SYCP2 overexpression does not change the RPA foci.

4. The authors concluded that SYCP2-dependent HR confers resistance to DNA-damaging agents in the abstract. However, the authors have used a reporter assay or IR throughout the manuscript. If the authors insist that the HR induced by DNA-damaging agents such as PARPi or TOP1i is enhanced by SYCP2 expression, at least, RPA and RAD51 foci following PARPi or TOP1i +/- SYCP2 should be examined.

Other comments and questions:

1. The authors mainly based their conclusion on the depletion of SYCP2 in HeLa cells or the overexpression of SYCP2 in U2OS cells (osteosarcoma) in the reporter assay. In Figure 1, is greater RAD51 foci observed in the tumor sample?
2. In Figure 2B, it is unusual to examine the induction of DSBs at 12 h post IR. To clarify the induction of DSBs post IR, the authors should count gH2AX foci at 1–2 h post IR. The number must be countable after 2 Gy.
3. In Figure 2D, the kinetics of protein recruitment by laser does not always represent the cascade of DSB repair because the laser induces DSBs, SSBs, and base damage. RPA foci should be examined after IR.
4. In Figure 3, the data regarding KR is convincing; however, it is unclear whether this result is related to cell toxicity by PARPi or TOP1i. Why didn't the authors simply examine S9.6 signal after PARPi or TOP1i exposure?
5. In Figure 4B and F, quantification is required. This is particularly important in Figure 4F, since the signal of the R-loop is too weak to determine the formation quantitatively. A negative control to show that a protein does not bind to the R-loop is required.
6. In Figure 4C, the second lane from the left may be - (SYCP2) and + (LMNA-CAS9)?
7. The data interpretation in Figures 6D–E is a bit confusing. To clarify that SYCP2 and BRCA1 are independently required for RAD51 foci formation, the authors should examine RAD51 foci in HeLa cells +/- siSYCP2 +/- siBRCA1.
8. Throughout the study, the authors used different DNA damage sources in each figure. In Figure 6E, why did the authors use CPT11? This should be also done by PARPi and Cisplatin to strengthen the statement in the abstract.
9. BRCA1 and 53BP1 gene status in MDA-MB-231 should be mentioned in the text. Is SYCP2 required for HR in BRCA1+53BP1 double-depleted cells? This is important when drug resistance is discussed.
10. In page 13, a reference should be inserted in "(ref)".
11. In the legend for Figure 2, what is the meaning of "4 U2OS cells"?
12. Does the overexpression of SYCP2 in non-malignant cells such as fibroblasts show a similar phenotype, i.e. are the findings in this study cancer-specific?
13. Surprisingly, SYCP2 alone, but not in a complex, was sufficient to promote HR. Did the authors check whether SYCP2 expression is associated with SYCP1 and SYCP3 in tumor samples? This should be discussed.

Reviewer #3 (Remarks to the Author):

The manuscript written by Wang et al reports a testis-specific protein SYCP2 functioning in RAD51-mediated homologous recombination, through which it contributes to therapy resistance in cancer treatment. The authors initially identified SYCP2 is aberrantly overexpressed in cancer types including breast and ovarian cancer. Then they revealed that SYCP2 is strongly associated with resistance to DDR drugs including PARPi. Mechanistically, they proposed SYCP2 stimulates TC-HR by promoting R-loop formation and DSB-attraction of RAD51 but independent of BRCA1/2. Finally, they came back to

the relevance of SYCP2 in cancer therapy by showing its high expression contributes to poor prognosis and resistance to TOP1 inhibitor and platinum. They conclude that SYCP2 is a promising biomarker to predict the resistance to DDR drugs.

The overall discovery of the aberrant SYCP2 expression in cancers is sound and significant, supported with combined evidence of biological and clinical assays. Multiple cellular and biochemical methodologies were exploited in this study to support their claims. The writing and readability of the manuscript is largely acceptable.

However, the SYCP2-dependent regulation on RAD51 and R-loop, as well as its relationship to BRCA1, remain inadequately clarified. The contribution of these mechanisms to drug resistance are mainly speculative. The work is also occasionally weakened by data quality and methodology defects in figures listed below:

1. Apparently, SYCP2 has more functions than transcription-coupled repair. It is important to consider the role of SYCP2 in drug resistance in the context of IR-induced and transcription-coupled RAD51 loading, as well as its BRCA1-dependent and independent functions. In the case of IR, RAD51 function that requires SYCP2 should be BRCA1/BRCA2 dependent.

2. In several experiments, IR experiments were used for analyzing SYCP2 functions in DSB repair (Fig. 2, 6D). It is still possible to address its role in IR-induced DSB repair in the context of TC-HR and R-loop. For example, inhibitors that suppress POL2 activity (amanitin, BRD4 inhibitors), as well as siRNA for RNase H, could be used to show if IR-induced RAD51 foci is reduced upon transcription inhibition.

3. Fig. 4C. again, I am not convinced that the repair assay for DSBs induced in CRISPR-based mClover HR assay and those induced in DART assay reflect the same mechanism. Despite authors claimed that Lamin A is actively transcribed, seemingly it is similar to the situation of reporter in Fig. 3A, the CRISPR-induced DSBs are nevertheless 'straight' DSBs and more likely resemble the IR situation. This system may not involve TC-hR and R-loop at all. In contrast, the DNA metabolism in DART system is complicated by the break processing of SSB into DSB by moving RNA polymerase, whose metabolism is quite different from straight DSBs. May the authors clarify this as well.

4. Fig. 5. The assays for requirement of KR-rich motif for R-loop and TC-HR is convincing. It is worth of investigating the impact of mutants on IR-induced RAD51. Importantly, it is conceivable that the mutants would abolish R-loop binding. By direct comparing wt-mutant activities for R-loop affinity would strengthen the in vitro assay in Fig 4E.

5. Line 287, authors realize that observations in Fig. 6E do not exclude the possibility that SYCP2 facilitates BRCA-mediated HR when BRCA1/2 are present. This can be tested using canonical I-Sce1 assay.

6. Fig. 1C, Fig. 7A and Suppl Fig. 5C. Poor image quality and it appears that SYCP2 positivity are cytoplasmic rather than nuclear, where it should be enriched to mediate HR, despite that aberrant overexpression in cancer cells often causes abnormal subcellular localization.

7. One of the major problems in this manuscript is the uncoupling of mechanistic study and clinical relevance. For example, I cannot see clearly how TC-HR and SYCP2-assisted RAD51 load to DSB contributes to resistance to DDR drug resistance. Does the BRCA1-dependent function of SYCP2, or the role of SYCP2 in replication also account for the resistance?

8. Supplementary Figure 5, the SYCP2 overexpression does not correlate with various markers of tumor progression. As for its important function in TC-HR, it is helpful to explore the correlation with status of sex hormone receptor (ie. ER levels or outcome of hormone treatment (TAM)).

9. As stated above, in Fig. 4E, 7A as well as other data, siSYCP2 was used to demonstrate the curbing

of cell/tumor growth in combination with PARPi or CPT. However, none of the experiment is shown using SYCP2 overexpression to directly reveal its correlation to drug resistance. Speculatively, overexpression of this protein in SYCP2-low cells may increase cell viability upon PARPi/TOPI treatments. Another genetic method for drug resistance study is to screen resistant clones and characterize the clonal phenotypes (ie. SYCP2 induction in comparison to the parental clone).

Minor comments:

1.U2OS is the major tool cell in the work, better to provide basic characterization like those in Figure 2A and Supplementary Figure 2A

2.Fig. 4C. the assay using siRNA-resistant plasmid is better choice.

3.Fig. 6E. As for TOP1i, PARPi is worth of investigating.

4.Typos including Line 281, TOPis, MDAMB231 in several places,

5.Fig. 6D. RAD51 foci counting in 20 cells is not statistically sufficient. Normally count for >200 cells.

Also, the cell identity in 2 graphs were not specified.

6.Fig. 7A. It is worth of showing body weight curve, as the therapy outcome may be complicated by toxicity caused by off-target effect on host mice that also curbs tumor progression.

7.Supplementary 4D, count IRIF number rather than %positivity as that in Fig. 6D.

Point by point responses to reviewer's comments.

Reviewer #1:

I think the study is rich of data both in vitro and in vivo, is well written and clear, and definitely relevant for cancer treatment. Experiments are generally well designed, and conclusion are mostly agreeable. Therefore, I moderately support publication in Nature Communication after a step of revision.

We sincerely appreciate the reviewer for their positive comments. We have carefully reviewed the reviewer' concerns and performed experiments as outlined below.

That is because, at this stage, I believe two important pieces of information are missing: 1) what recruits SYCP2 to site of DNA damage in cancer cells (are DNA damage sensors or transcription itself? Chromatin modification? H3K36 was previously suggested to stimulate HR for transcribed genes by Legube laboratory...); is MRN complex involved?

Thanks for reviewer's suggestion. To investigate the *impact of transcription* on the recruitment of SYCP2, we examined the effect of RNA polymerase II activity on the recruitment of SYCP2. Cells were treated with the RNA polymerase II inhibitor α -amanitin or DRB, and we observed that neither DRB nor α -amanitin affected the recruitment of SYCP2 to the TA-KR sites (**Supplementary Fig. 4A**). We further investigated the impact of DNA damage sensors and chromatin states on the recruitment of SYCP2. It is well-established that ATM is required for DNA damage-induced transcription repression¹. Additionally, previous studies have demonstrated that H3K36 trimethylation is involved in recruiting CtIP, facilitating resection and HR at active genes². In yeast, the conserved chromatin remodeler ISWI, which contains the enzymatic subunit SMARCA5, plays a role in maintaining H3K36 methylation^{3, 4}; To understand *the effects of DNA damage sensors and chromatin states* on SYCP2 recruitment, we employed inhibitors targeting ATM, ATR, and DNAPK, as well as siRNA against CtIP, Mre11 (another major end processing enzyme of DSBs), and SMARCA5. Interestingly, none of ATMi, ATRi, DNAPKi, siMre11 or siCtIP showed significant effects on SYCP2 recruitment (**Supplementary Fig. 4B & 4C**). However, a notable reduction in SYCP2 foci at TA-KR sites was observed in the siSMARCA5 group compared to the siCtrl group (**Figure attached**). These findings suggest that SMARCA5 may serve as one of the upstream regulators of SYCP2 recruitment. Given this manuscript focuses on the function of SYCP2 in HR, we will conduct future studies to explore the mechanisms of SYCP2 recruitment by H3K36 methylation and SMARCA5.

U2OS-TRE cells transfected with TA-KR and siSMARCA5 or siCtrl were light-activated, recovered for 30 min, fixed, and stained with anti-SYCP2. Fold increase of SYCP2 foci at sites of KR compared to background was quantified (n=30, mean \pm SD). Experiments were repeated 3 times, images from one representative experiment were shown.

2) In addition, it is not clear which is the mechanism by which SYCP2 might promote DNA:RNA hybrids formation and HR. Could it be that SYCP2 controls HR and hybrid formation by supporting resection? What about CtIP/BRCA1/pRPA recruitment after KD of SYCP2? Is the effect on RAD51 foci due to less resection? They should test efficacy of resection, for example by BrdU native staining after IR.

We thank the reviewer bringing out this question for further discussion. To investigate whether the resection efficiency affect hybrids formation and repair, we utilized the AID-DivA reporter system developed by Dr. Gaëlle Legube's lab⁵. Remarkably, SYCP2 knockdown did not reduce end resection efficiency at DSB (**Supplementary Figure 2F**). Moreover, number of ionizing radiation-induced foci (IRIF) of CtIP/pRPA and SYCP2 are not influenced by each other (**Supplementary Figure 2G&H**). siBRCA1 did not influence the recruitment of SYCP2 and vice versa (**Figure 6B**). Together, we propose that function of SYCP2 in repair does not affect and is not affected by end resection process. The mechanism by which SYCP2 promotes hybrids formation is mediated by its DNA damage response and hybrids binding affinity of M1-domain of SYCP2 (**Figure 4B-E**). As discussed above, this activity might also be promoted by SMARCA5-mediated chromatin remodeling after damage.

Other specific points

Authors show that SYCP2 KD reduces HR by DR-GFP reporter system while SYCP2 overexpression enhances it. Would be interesting to test also the impact of SYCP2 KD or its overexpression on NHEJ and alternative NHEJ with EJ5-GFP and EJ2-GFP respectively? Are other repair pathways affected as well?

We thank the reviewer for the suggestions. In response to the comment, we performed experiments to assess the impact of SYCP2 knockdown and overexpression on the frequency of non-homologous end joining (NHEJ) and alternative non-homologous end joining (alt-NHEJ) using EJ5-GFP and EJ2-GFP reporter cells, respectively (**Supplementary Figure 2E**). We did not observe significant changes of repair products in both reporter assays upon knocking down or overexpressing (OE) SYCP2. These results indicate that SYCP2 does not have a substantial influence on the occurrence of these DNA repair pathways.

Figure 2E: how does the undamaged cells look-like? Please show the undamaged cells for both siCTRL/siSYCP2.

We appreciate the reviewer's suggestion, and as suggested, we measured RAD51 foci in undamaged cells in siCtrl and siSYCP2 treated cells. RAD51 did not form foci without exogenous damage in siCtrl or siSYCP2 treated cells (**Supplementary Figure 2I**).

Are the indicated cells in S/G2? Please use a co-staining with Cyclin A and RAD51 to confirm it both in Figures 2 and Supplementary Figure 2C and D.

To investigate the potential cell cycle dependency of SYCP2 function, we performed co-staining of Cyclin A and RAD51. As the reviewer indicated, RAD51 IR-induced foci were observed in Cyclin A-positive cells in U2OS (left) and MDA-MB-231 (right) cells (**Supplementary Figure 3C**).

In the manuscript, data demonstrate that transcription inhibition reduces HR. Nevertheless, a recent paper suggests the opposite, showing that inhibition of transcription increases resection and HR ([https://www.cell.com/cell-reports/pdfExtended/S2211-1247\(22\)00277-7](https://www.cell.com/cell-reports/pdfExtended/S2211-1247(22)00277-7)). This paper should be mentioned and commented to highlight incongruencies.

As the reviewer suggested, we now added this paper into our discussion. This study published in Cell Reports has highlighted the role of BMI-1-dependent transcriptional inhibition in promoting DNA end resection and HR. Following DNA damage, there is a temporary inhibition and repression of transcription to facilitate repair processes. In our previous studies, we have published evidence showing that temporary transcription repression induced by damage is associated with efficient R-loop-dependent repair⁶⁻⁹. Therefore, the repression of transcription induced by DNA damage promotes homologous recombination (HR). When we refer to "transcription inhibition," we mean the blocking of transcription, which results in the absence of transcription events. Consequently, repair processes that rely on transcription, such as transcription-coupled repair (TC-HR), will be impeded. Hence, the concept that "DNA damage-induced transcription repression promotes TC-HR" and "transcription-inhibition mediated TC-HR inhibition" are not contradictory. They are complementary and highlight the intricate relationship between DNA damage, transcription, and repair mechanisms.

The authors show that SYPC2 is recruited to ROS induced DNA damage in a transcription dependent manner. Is it occurring also at DSB induced by AsiSI restriction enzyme in the DivA cellular system that cut at active transcribed genes?

We examined the recruitment of SYCP2 at double-strand breaks (DSBs) induced by I-SCEI at the TA-Cherry sites using our DART system⁸. For this purpose, I-SCEI sites were inserted adjacent to the TRE array, which was integrated into the U2OS genome. We did observe that SYCP2 is recruited to enzyme induced DSBs (**Figure 3C**).

Figure 3D: SYPC2 is required for DNA:RNA hybrids formation at damaged sites as detected by S9.6 DRIP. DNA end resection, is functional to induce DNA:RNA hybrids formation upon damage (see <https://pubmed.ncbi.nlm.nih.gov/30560944/>). Does resection take place to allow hybrid establishment in the presence of SYPC2?

Our previous findings, as discussed in earlier responses, demonstrate that the function of SYCP2 is independent of end resection. Additionally, the level of R-loops before and after damage at sites of TA-KR remains unchanged regardless of the presence or absence of CtIP. These results suggest that SYCP2 operates through mechanisms that are distinct from end resection. (**Supplementary Figure 3G**).

Which is the level of DNA:RNA hybrids in transcription OFF conditions as detected by S9.6 antibody both in the presence or in the absence of SYPC2?

R-loops are not detected when transcription is off in the absence or presence of SYPC2 and as shown in **Supplementary Figure 3F**.

In this study: <https://elifesciences.org/articles/69881>, it was instead shown that hybrids inhibits HR. I think it should be discussed.

We add following discussion in our manuscript: Although DNA-RNA hybrids inhibit canonical HR¹⁰, accumulating evidence show that DNA-RNA hybrids and R-loop could trigger TC-HR, in which DNA-RNA hybrids recruit DNA repair proteins differently from canonical HR¹¹⁻¹³.

Figure 4B a single image might not be sufficient to state which SYPC2 fragment is required for efficient recruitment. A simple quantification with the percentage of cell might be added.

The quantification of both the frequency and fold increase of mean intensity has been added to the **Figure 4B**.

In Figure 5 D and E it should be shown also the transcription OFF condition, at least for the control sample with endogenous SYPC2.

There is no recruitment of endogenous SYCP2 when transcription is off. The frequency and intensity of endogenous SYCP2/RAD51 foci under transcription-off conditions are shown in **Supplementary 5B and 5C**.

Figure 6: it shows that SYPC2 acts independently from BRCA1/2 status. Is SYPC2 overexpressed in the cancer cell lines tested (HCC1937 and HCC1954)? Maybe the lack of activity of BRCA1/2 induces SYPC2 hyper activation to compensate lack of HR. It would be also important to check by western blotting if SYPC2 KD alters or not BRCA1/2 levels.

We appreciate the suggestion provided by the reviewer. We conducted experiments involving the overexpression of SYCP2 in both HCC1937 and HCC1954 cell lines, which resulted in a significant increase in the number and frequency of RAD51 IR-induced foci that colocalized with γ -H2A.X two hours after a 2 Gy IR treatment. This observation suggests that the elevated levels of SYCP2 promote the formation of RAD51 foci in BRCA1/2 deficient cells (**Figure 6E**). Moreover, SYCP2 knockdown did not affect the expression of BRCA1/2 in the tested cell lines (**Figure 6D**).

Line 294 a reference is missing "(ref)"?

Thank you for bringing this to our attention. We have included the appropriate reference in the manuscript to support our statement.

Reviewer #2 (Remarks to the Author):

We thank for the reviewer's summary, and we are very delighted that the reviewer finds our finding of the correlation between high SYCP2 levels and drug resistance is important. We performed experiments and discussed the reviewer's suggestions.

1. One of the critical points that lead to confusion is the mechanism by which SYCP2 expression enhances HR in mitotic cells. The authors proposed a model in which SYCP2 overexpression enhances the generation of the R-loop at the DSB site in a transcription-dependent manner. If the formation of the R-loop is dependent on ongoing transcription, the increase in R-loop of SYCP2 overexpression cells may be caused by the upregulation of transcription events. Does SYCP2 overexpression upregulate the number of transcription events per cell, which increases the opportunity for DSB and transcription collision? Has the overall transcriptional activity been examined in SYCP2-expressing cells?

To assess whether SYCP2 overexpression impacted overall transcriptional levels, we quantified the mRNA levels in cells with or without SYCP2 overexpression. We observed that the mRNA levels were significantly lower in the SYCP2 knockdown group compared to the siCtrl group. Conversely, overexpression of SYCP2 increased the mRNA levels compared to the vehicle control. These findings suggest that increased SYCP2 levels upregulate the overall transcriptional activities of the cells, supporting the reviewer's speculation that SYCP2 overexpression promotes DSB-transcriptional collisions by upregulating transcriptional events (**Supplementary Figure 8B**).

2. Another critical question is whether enhanced HR maintains the fidelity of DSB repair. For example, RAP80 depletion enhances DSBs undergoing resection but confers error-prone repair. The authors should examine genome stability following SYCP2-dependent HR to clarify whether increased cell viability (drug resistance) is associated with less genome instability, e.g., by chromosomal aberration assay.

We appreciate the valuable input from the reviewer. In response to their suggestions, we performed centromere and DAPI staining of the chromosomes during metaphase. Interestingly, we observed increased rate of chromosomal aberrations in siSYCP2 treated cells and a slightly lower level of chromosomal aberrations in the SYCP2-overexpressing HeLa cells compared to the vehicle group (**Supplementary Figure 8C**). These results support the notion that SYCP2 may play a role in maintaining chromosomal stability and integrity.

3. In the model, the authors proposed that SYCP2 promotes R-loop formation at TC-DSB sites without enhancing resection. However, the authors did not examine RPA IRIF. The authors should confirm that SYCP2 overexpression does not change the RPA foci.

We thank the reviewer's comments. We examined the formation of IR-induced RPA foci in cells overexpressing SYCP2 compared to the vehicle control. Comparable level of RPA foci was observed in both groups, confirming that SYCP2 overexpression did not affect the formation of RPA foci (**Supplementary Figure 3D**). These findings suggest that SYCP2 does not directly influence the recruitment or dynamics of RPA during the DNA damage response.

4. The authors concluded that SYCP2-dependent HR confers resistance to DNA-

damaging agents in the abstract. However, the authors have used a reporter assay or IR throughout the manuscript. If the authors insist that the HR induced by DNA-damaging agents such as PARPi or TOP1i is enhanced by SYCP2 expression, at least, RPA and RAD51 foci following PARPi or TOP1i +/- SYCP2 should be examined.

We thank the reviewer's comments. In response, we conducted an analysis of RPA/RAD51 foci following PARPi/TOP1i treatment in U2OS cells. **Supplementary Figure 3E** illustrates our findings, demonstrating that siSYCP2 treatment resulted in a reduction of RAD51 foci (upper panel), while having no discernible impact on RPA foci (lower panel).

Other comments and questions:

1. The authors mainly based their conclusion on the depletion of SYCP2 in Hela cells or the overexpression of SYCP2 in U2OS cells (osteosarcoma) in the reporter assay. In Figure 1, is greater RAD51 foci observed in the tumor sample?

RAD51 foci can be detected in cells but cannot be detected in IHC in tumor samples.

Moreover, RAD51 expression was not affected in siSYCP2 in U2OS cells (**Fig. 2E**) and two breast cancer cell lines (**Fig. 6D**).

2. In Figure 2B, it is unusual to examine the induction of DSBs at 12 h post IR. To clarify the induction of DSBs post IR, the authors should count gH2AX foci at 1–2 h post IR. The number must be countable after 2 Gy.

We have examined the gH2A.X foci following a 2 Gy IR treatment at 2 hours. At this time point, the gH2A.X foci were clearly observable and quantifiable. The updated graph displaying this information is presented in **Figure 2B and Supplementary Figure 2C**.

We observed no significant difference in the foci numbers between the SYCP2 knockdown and control groups at 2 hours. However, we did detect a delayed clearance of gH2A.X foci at 24 and 48 hours post 2 Gy IR treatment, suggesting the involvement of SYCP2 in DNA double-strand break repair.

3. RPA foci should be examined after IR.

We counted the RPA foci induced by 2 Gy IR at 2 hours. The foci number showed no significant changes in Vehicle and SYCP2 overexpressed cells (**Supplementary Fig. 3D**), suggesting the recruitment of RPA is not dependent on the expression of SYCP2.

4. In Figure 3, the data regarding KR is convincing; however, it is unclear whether this result is related to cell toxicity by PARPi or TOP1i. Why didn't the authors simply examine S9.6 signal after PARPi or TOP1i exposure?

Thanks for reviewer's suggestion. We did observe decreased Pan-S9.6 signal following the treatment of PARPi or TOP1i, indicating SYCP2-dependent R-loop formation contributes to repair (**Supplementary Fig. 6G**).

5. In Figure 4B and F, quantification is required. This is particularly important in Figure 4F, since the signal of the R-loop is too weak to determine the formation quantitatively. A negative control to show that a protein does not bind to the R-loop is required.

The quantification of 4B and 4F were added to the main Figure. Given there is no previous studies clearly proved any proteins which are not required for R-loop formation, in addition to R-loop formation assay, we conducted an EMSA assay. We examined the hybrids binding with M1 and M1-2KR&5KR mutants. M1 and M1-2KR exhibited higher affinity to the substrates, whereas M1-5KR did not show a specific binding activity (**Supplementary Figure 5E**), showing the direct affinity with DNA-RNA hybrids of M1 domain of SYCP2.

6. In Figure 4C, the second lane from the left may be – (SYCP2) and + (LMNA-CAS9)? Thanks for pointing it out. Yes, we have corrected the label of Figure 4C.

7. The data interpretation in Figures 6D–E is a bit confusing. To clarify that SYCP2 and BRCA1 are independently required for RAD51 foci formation, the authors should examine RAD51 foci in HeLa cells +/- siSYCP2 +/- siBRCA1.

We thank the reviewer's comments. We examined the RAD51 IRIF in HeLa cells with single knockdown of SYCP2 or BRCA1 and double knockdown of both SYCP2 and BRCA1 compared to control. As expected, single KD of either SYCP2 and BRCA1 or Double KD all attenuates the RAD51 IRIF (**Supplementary Figure 6E**). Importantly, double KD further reduced of RAD51 foci formation compared to each of the single KD.

8. Throughout the study, the authors used different DNA damage sources in each figure. In Figure 6E, why did the authors use CPT11? This should be also done by PARPi and Cisplatin to strengthen the statement in the abstract.

Thanks for the reviewer's suggestion. We performed the new survival assay using PARPi and Cisplatin in BRCA1/SYCP2 single knockdown and double knockdown cells. In both survival assays, double knockdown of BRCA1 and SYCP2 make the cells more sensitive to PARPi and Cisplatin compared to the single knockdown of BRCA1 or SYCP2, consistently with our previous survival result using the TOP1i (CPT11) treatment (**Supplementary Figure 6F**).

9. BRCA1 and 53BP1 gene status in MDA-MB-231 should be mentioned in the text. Is SYCP2 required for HR in BRCA1+53BP1 double-depleted cells? This is important when drug resistance is discussed.

We thank the reviewer's comments. BRCA1 or 53BP1 are normally expressed and do have mutations in MDA-MB-231 cell line. We performed the experiments using BRCA1+53BP1 double-depleted cells and added the discussion. SYCP2 is required for HR in BRCA+53BP1 double KD cells (**Supplementary Fig. 8D left**). Both in the presence and absence of 53BP1, overexpression of SYCP2 contributes to increased HR activity (**Supplementary Fig. 8D right**), suggesting that SYCP2 contributes to drug resistance independently of both BRCA1 and 53BP1.

10. In page 13, a reference should be inserted in "(ref)". Reference is added back to the text.

11. In the legend for Figure 2, what is the meaning of "4 U2OS cells"?

We apologize for any confusion caused by the mis-inserted words. The figure legend for Figure 2 has been revised to accurately reflect the content and findings of the figure.

12. Does the overexpression of SYCP2 in non-malignant cells such as fibroblasts show a similar phenotype, i.e. are the findings in this study cancer-specific?

We thank the reviewer for pointing this out. In order to investigate the impact of SYCP2 in non-malignant cells, we conducted a DR-GFP HR assay by overexpressing SYCP2 in normal breast cells. The relative HR efficiency increased after the overexpression of SYCP2 in normal MCF10A cells, indicating that the HR function of SYCP2 can be enhanced solely by its overexpression in non-malignant cells (**Figure 2C**).

13. Surprisingly, SYCP2 alone, but not in a complex, was sufficient to promote HR. Did the authors check whether SYCP2 expression is associated with SYCP1 and SYCP3 in tumor samples? This should be discussed.

We conducted a comprehensive bioinformatic analysis utilizing the TCGA database, comprising a sample size of 1215. Our analysis revealed that among the SYC family genes, SYCP2 is the only gene that showed significant upregulation in breast cancer compared to normal breast samples. This finding is illustrated in **Supplementary Figure 1D**. While SYCP1 and SYCP3 share certain structural and domain similarities with SYCP2, we showed the C-terminus of SYCP2, which binds to SYCP3, is not required for its role in R-loop dependent repair. SYCP2 possesses a unique domain showed in Figure 4 that potentially serves distinct functions in cancer cell survival and DNA damage repair.

Reviewer 3

The overall discovery of the aberrant SYCP2 expression in cancers is sound and significant, supported with combined evidence of biological and clinical assays. Multiple cellular and biochemical methodologies were exploited in this study to support their claims. The writing and readability of the manuscript is largely acceptable...The work is also occasionally weakened by data quality and methodology defects in figures listed below:

1. Apparently, SYCP2 has more functions than transcription-coupled repair. It is important to consider the role of SYCP2 in drug resistance in the context of IR-induced and transcription-coupled RAD51 loading, as well as its BRCA1-dependent and independent functions. In the case of IR, RAD51 function that requires SYCP2 should be BRCA1/BRCA2 dependent.

We appreciate the reviewer's comments. We completely agree with the reviewer regarding the diverse mechanisms of SYCP2's function in drug resistance, particularly in the context of IR-induced and transcription-coupled recombinational repair, especially when considering the presence or absence of BRCA1. Our results suggest that in cells with wildtype BRCA1, SYCP2 acts as a backup to facilitate BRCA1-dependent loading of RAD51, thereby promoting the repair of IR-induced DSBs for repair IR-induced DSBs. However, in the absence of BRCA1, SYCP2 assumes a critical role in RAD51 loading, particularly when damage occurs at transcribed regions of the genome. SYCP2 can independently facilitate RAD51 loading at sites of R-loops, underscoring its predominant function in such scenarios. The data are presented and discussed below.

2. In several experiments, IR experiments were used for analyzing SYCP2 functions in DSB repair (Fig. 2, 6D). It is still possible to address its role in IR-induced DSB repair in the context of TC-HR and R-loop. For example, inhibitors that suppress POL2 activity (amanitin, BRD4 inhibitors), as well as siRNA for RNase H, could be used to show if IR-induced RAD51 foci is reduced upon transcription inhibition.

We thank the reviewer's suggestion. We performed RAD51 IRIF with the treatment of suggested POLII inhibitors includes DRB, α -amanitin and siRNase H1. To our surprise, cells only showed increased RAD51 foci numbers with overexpressed SYCP2 without any treatment of PolII inhibitors. In cells with treatment of PolII inhibitors, RAD51 IR induced foci numbers have not significantly increased after overexpressing SYCP2. Similar trends have exhibited in the siRNase H1 group, no significant change of RAD51 foci numbers detected in the overexpressed group (Figure attached) .

The numbers of RAD51 IRIF in siCtrl and siSYCP2 U2OS cells with treatment of DMSO or 20 μ M DRB or 100 μ g/ml α -amanitin or siRNaseH1 1 hr after 2 Gy IR were quantified (n=200, +/- SD). Experiments were repeated 3 times, the representative images of RAD51 from one experiment in siCtrl and siSYCP2 U2OS cells with indicated treatment were shown on the left. Statistical analysis was done with the Student-t-test, ***: p<0.0001.

3. Fig. 4C. again, I am not convinced that the repair assay for DSBs induced in CRISPR-based mClover HR assay and those induced in DART assay reflect the same mechanism. Despite authors claimed that Lamin A is actively transcribed, seemingly it is similar to the situation of reporter in Fig. 3A, the CRISPR-induced DSBs are nevertheless 'straight' DSBs and more likely resemble the IR situation. This system may not involve TC-hR and R-loop at all. In contrast, the DNA metabolism in DART system is complicated by the break processing of SSB into DSB by moving RNA polymerase, whose metabolism is quite different from straight DSBs. May the authors clarify this as well.

The reviewer's observation is accurate. In our study, we utilized CRISPR-based mClover as a means to validate the results obtained from the DR-GFP reporter assay. Moreover, we recently showed that RNA transcripts significantly stimulate HR by forming R-loops at the transcribed region of the genome⁷. We established a Tet-DR-GFP HR assay, which a modified DR-GFP HR assay to compare the HR activities in transcriptionally on and off states⁷. As we reported, HR efficacy was about 2-fold higher when transcription is on in cells⁷ (**Fig. 3A**). Interestingly, SYCP2 KD reduced HR activity when transcription was on, but did not further reduce HR when transcription was off (**Fig. 3A**), suggesting that SYCP2 promotes HR in a transcription-dependent manner. Therefore, results from the Tet-DR-GFP reporter assay (**Fig. 3A and Fig. 5G**) demonstrated the dependence of homologous recombination (HR) activities on transcription.

We acknowledge and agree with the reviewer's perspective on the distinct manner in which the DART system generates double-strand breaks (DSBs) compared to enzyme-induced DSBs. Importantly, we emphasize that the DART system closely mimics the cellular response observed in vivo, making it a valuable tool for our research. Furthermore, in our revised manuscript, we included an investigation into the recruitment of SYCP2 at I-SCEI induced DSBs in the transcribed regions of the genome. As depicted in **Figure 3C**, we show that SYCP2 is also recruited to I-SCEI

induced DSBs in transcribed regions of the genome. We have included these findings and corresponding comments in our revised manuscript.

4. Fig. 5. The assays for requirement of KR-rich motif for R-loop and TC-HR is convincing. It is worth of investigating the impact of mutants on IR-induced RAD51. Importantly, it is conceivable that the mutants would abolish R-loop binding. By direct comparing wt-mutant activities for R-loop affinity would strengthen the in vitro assay in Fig 4E.

As the reviewer suggested, we investigated recruitment of each KR mutants after IR and found M1-2KR but not M1-5KR form IRIF (**Fig. 5E left**). M1-2KR but not M1-5KR rescued IRIF of RAD51 (**Fig. 5E right**). We also conducted EMSA to test the direct binding between SYCP2-M1 to DNA:RNA hybrids. We also found that M1-2KR but not M1-5KR has the affinity with hybrids as compared to the M1 domain (**Supplementary Fig. 5E**).

5. Line 287, authors realize that observations in Fig. 6E do not exclude the possibility that SYCP2 facilitates BRCA-mediated HR when BRCA1/2 are present. This can be tested using canonical I-Sce1 assay.

We have used I-SCEI to induce DSB in the DR-GFP reporter assay. Cells with BRCA1 and SYCP2 knockdown showed significant decrease of HR efficiency, while the double knockdown of both SYCP2 and BRCA1 further attenuates the HR efficiency (**Supplementary Figure 8D**), suggesting SYCP2 may facilitate HR in the absence of BRCA1.

6. Fig. 1C, Fig. 7A and Suppl Fig. 5C. Poor image quality and it appears that SYCP2 positivity are cytoplasmic rather than nuclear, where it should be enriched to mediate HR, despite that aberrant overexpression in cancer cells often causes abnormal subcellular localization.

We have changed our images to high quality figures. The unclear localization of SYCP2 is possibly due to low resolution of figures, we enlarged part of the figure to represent the localization of SYCP2.

7. One of the major problems in this manuscript is the uncoupling of mechanistic study and clinical relevance. For example, I cannot see clearly how TC-HR and SYCP2-assisted RAD51 load to DSB contributes to resistance to DDR drug resistance. Does the BRCA1-dependent function of SYCP2, or the role of SYCP2 in replication also account for the resistance?

Our previous studies suggest that damage-induced R-loops stimulate RAD51 recruitment⁸ and RAD51-mediated D-loop formation⁷, which may explain the positive effects of SYCP2 on HR (**Fig. 7D**). In addition to its role in loading RAD51 to DSBs, we found that SYCP2 is also required for protecting nascent DNA from degradation at stalled replication forks (**Supplementary Fig. 8A**), which is known to be a RAD51-dependent process. Recent studies by others and us suggested that the degradation of nascent DNA at stalled forks and single-stranded DNA (ssDNA) gaps generated at stalled forks may contribute to the PARPi sensitivity of BRCA-deficient cells¹⁴⁻²⁰. It is possible that SYCP2 confers PARPi resistance by promoting RAD51 loading to stalled forks or ssDNA gaps, providing an attractive hypothesis to test in future studies.

8. Supplementary Figure 5, the SYCP2 overexpression does not correlate with various markers of tumor progression. As for its important function in TC-HR, it is helpful to explore the correlation with status of sex hormone receptor (ie. ER levels or outcome of hormone treatment (TAM)).

We thank the reviewer's question and we have done the correlation analysis between the status of ER levels with SYCP2 expression and the outcome of hormone treatment (Tamoxifen) with SYCP2 expression. There is no obvious correlation between ER status and SYCP2 expression levels (**Supplementary 7B**). Similarly, there is also no significant correlation between the outcomes of Tamoxifen treatment and SYCP2 expression (**Supplementary 7C**), indicating SYCP2-dependent repair is ER-independent pathway.

9. As stated above, in Fig. 4E, 7A as well as other data, siSYCP2 was used to demonstrate the curbing of cell/tumor growth in combination with PARPi or CPT. However, none of the experiment is shown using SYCP2 overexpression to directly reveal its correlation to drug resistance. Speculatively, overexpression of this protein in SYCP2-low cells may increase cell viability upon PARPi/TOPI treatments. Another genetic method for drug resistance study is to screen resistant clones and characterize the clonal phenotypes (ie. SYCP2 induction in comparison to the parental clone).

Thanks for the reviewer's suggestion. We performed the new survival assay using PARPi and Cisplatin in BRCA1/SYCP2 single knockdown and double knockdown cells. In both survival assays, double knockdown of BRCA1 and SYCP2 make the cells more sensitive to PARPi and Cisplatin compared to the single knockdown of BRCA1 or SYCP2, consistently with our previous survival result using the TOP1i (CPT11) treatment (**Supplementary Figure 6F**).

Minor comments:

1. U2OS is the major tool cell in the work, better to provide basic characterization like those in Figure 2A and Supplementary Figure 2A

We added the survival graph of U2OS cells in **Supplementary Figure 2B**.

2. Fig. 4C. the assay using siRNA-resistant plasmid is better choice.

In Fig. 4C, we overexpressed SYCP2. Rescue experiment using WT and mutants of SYCP2 in the same reporter assay with siSYCP2 is shown in **Fig. 5F**. In **Fig. 5D-5H**, we performed several rescue experiments in siSYCP2 pre-knocked down cells with siSYCP2^{#2}. Our plasmids are resistant to siSYCP2^{#2}.

3. Fig. 6E. As for TOP1i, PARPi is worth of investigating.

Thanks for reviewer's suggestion. We have added a survival graph of PARPi and cisplatin in **Supplementary Figure 6F**.

4. Typos including Line 281, TOPis, MDAMB231 in several places,

We thank the reviewer for pointing this out and we have corrected the typos throughout the paper.

5.Fig. 6D. RAD51 foci counting in 20 cells is not statistically sufficient. Normally count for >200 cells. Also, the cell identity in 2 graphs were not specified.

We have corrected the graph by counting foci in more than 200 cells.

6.Fig. 7A. It is worth of showing body weight curve, as the therapy outcome may be complicated by toxicity caused by off-target effect on host mice that also curbs tumor progression.

We have included a graph displaying the body weight of mice before and after treatment in four groups. No significant alterations in body weight were observed across the treated groups (**Supplementary Figure 7A**)

7.Supplementary 4D, count IRIF number rather than %positivity as that in Fig. 6D. IRIF number has been added into **Supplementary 6D**.

1. Sordet, O., Nakamura, A.J., Redon, C.E. & Pommier, Y. DNA double-strand breaks and ATM activation by transcription-blocking DNA lesions. *Cell Cycle* **9**, 274-278 (2010).
2. Pfister, S.X. *et al.* SETD2-dependent histone H3K36 trimethylation is required for homologous recombination repair and genome stability. *Cell Rep* **7**, 2006-2018 (2014).
3. Aydin, O.Z. *et al.* Human ISWI complexes are targeted by SMARCA5 ATPase and SLIDE domains to help resolve lesion-stalled transcription. *Nucleic Acids Res* **42**, 8473-8485 (2014).
4. Li, J. *et al.* H3K36 methylation and DNA-binding both promote loc4 recruitment and Isw1b remodeler function. *Nucleic Acids Res* **50**, 2549-2565 (2022).
5. Zhou, Y., Caron, P., Legube, G. & Paull, T.T. Quantitation of DNA double-strand break resection intermediates in human cells. *Nucleic Acids Res* **42**, e19 (2014).
6. Chen, H. *et al.* m(5)C modification of mRNA serves a DNA damage code to promote homologous recombination. *Nat Commun* **11**, 2834 (2020).
7. Ouyang, J. *et al.* RNA transcripts stimulate homologous recombination by forming DR-loops. *Nature* **594**, 283-288 (2021).
8. Teng, Y. *et al.* ROS-induced R loops trigger a transcription-coupled but BRCA1/2-independent homologous recombination pathway through CSB. *Nat Commun* **9**, 4115 (2018).
9. Wei, L. *et al.* DNA damage during the G0/G1 phase triggers RNA-templated, Cockayne syndrome B-dependent homologous recombination. *Proc Natl Acad Sci U S A* **112**, E3495-3504 (2015).
10. Ortega, P., Merida-Cerro, J.A., Rondon, A.G., Gomez-Gonzalez, B. & Aguilera, A. DNA-RNA hybrids at DSBs interfere with repair by homologous recombination. *Elife* **10** (2021).
11. Al-Hadid, Q. & Yang, Y. R-loop: an emerging regulator of chromatin dynamics. *Acta Biochim Biophys Sin (Shanghai)* **48**, 623-631 (2016).

12. Niehrs, C. & Luke, B. Regulatory R-loops as facilitators of gene expression and genome stability. *Nat Rev Mol Cell Biol* **21**, 167-178 (2020).
13. Petermann, E., Lan, L. & Zou, L. Sources, resolution and physiological relevance of R-loops and RNA-DNA hybrids. *Nat Rev Mol Cell Biol* **23**, 521-540 (2022).
14. Schlacher, K. *et al.* Double-strand break repair-independent role for BRCA2 in blocking stalled replication fork degradation by MRE11. *Cell* **145**, 529-542 (2011).
15. Ray Chaudhuri, A. *et al.* Replication fork stability confers chemoresistance in BRCA-deficient cells. *Nature* **535**, 382-387 (2016).
16. Yazinski, S.A. *et al.* ATR inhibition disrupts rewired homologous recombination and fork protection pathways in PARP inhibitor-resistant BRCA-deficient cancer cells. *Genes Dev* **31**, 318-332 (2017).
17. Cong, K. *et al.* Replication gaps are a key determinant of PARP inhibitor synthetic lethality with BRCA deficiency. *Mol Cell* **81**, 3128-3144 e3127 (2021).
18. Dias, M.P., Moser, S.C., Ganesan, S. & Jonkers, J. Understanding and overcoming resistance to PARP inhibitors in cancer therapy. *Nat Rev Clin Oncol* **18**, 773-791 (2021).
19. Simoneau, A., Xiong, R. & Zou, L. The trans cell cycle effects of PARP inhibitors underlie their selectivity toward BRCA1/2-deficient cells. *Genes Dev* **35**, 1271-1289 (2021).
20. Taglialatela, A. *et al.* REV1-Polzeta maintains the viability of homologous recombination-deficient cancer cells through mutagenic repair of PRIMPOL-dependent ssDNA gaps. *Mol Cell* **81**, 4008-4025 e4007 (2021).

REVIEWERS' COMMENTS

Reviewer #1 (Remarks to the Author):

In the revised version of the manuscript, the authors have experimentally addressed all my suggestion. They have included additional data supporting the role of SYCP2 in supporting HR by favoring R-loops accumulation at DSB occurring at transcribed loci, thus inducing resistance to PARP inhibitor in cancer.

In my opinion, they need to carefully check all the figures again since I see some inverted label in samples/staining (es. panel 3C staining are mislabeled)

For the rest the revised manuscript is now suited for publication in Nature Communications.

Reviewer #2 (Remarks to the Author):

The authors adequately answered all my concerns.

Reviewer #3 (Remarks to the Author):

The revised manuscript has been extensively improved and I am largely convinced that SYCP2 is an important factor in cancer resistance to genotoxic agents via R-loop mediated repair. The only thing I worry is the validity of their major claim of TC-HR throughout the manuscript, given the POL2 inhibitors (amanitin and BRD4i) did not impact the SYCP2-mediated RAD51-dependent repair. In the rebuttal for reviewer 1 and 3, the authors showed negative results without specific comments. It is worthwhile to discuss these in a more extensive way in the article to solidify the grounds that these SYCP2-mediated mechanistic events are transcription-dependent. In addition, despite it is negative result, the data for reviewer 3 question 2 should be included in the manuscript.

We sincerely appreciate the reviewer for their positive comments. We have carefully reviewed the reviewer's concerns and updated our manuscript.

Reviewer #1 (Remarks to the Author):

In the revised version of the manuscript, the authors have experimentally addressed all my suggestion. They have included additional data supporting the role of SYCP2 in supporting HR by favoring R-loops accumulation at DSB occurring at transcribed loci, thus inducing resistance to PARP inhibitor in cancer.

In my opinion, they need to carefully check all the figures again since I see some inverted label in samples/staining (es. panel 3C staining are mislabeled)

For the rest the revised manuscript is now suited for publication in Nature Communications.

We are very delighted that Reviewer #1 has been satisfied with our revised experimental answers. We appreciated the reviewer's suggestion regarding labels. We have corrected the label of panel 3C and reviewed all figures and legends ensuring accuracy and attention to detail as per the reviewer's advice.

Reviewer #2 (Remarks to the Author):

The authors adequately answered all my concerns.

We thank Reviewer #2's agreement for the review answer.

Reviewer #3 (Remarks to the Author):

The revised manuscript has been extensively improved and I am largely convinced that SYCP2 is an important factor in cancer resistance to genotoxic agents via R-loop mediated repair. The only thing I worry is the validity of their major claim of TC-HR throughout the manuscript, given the POL2 inhibitors (amanitin and BRD4i) did not impact the SYCP2-mediated RAD51-dependent repair. In the rebuttal for reviewer 1 and 3, the authors showed negative results without specific comments. It is worthwhile to discuss these in a more extensive way in the article to solidify the grounds that these SYCP2-mediated mechanistic events are transcription-dependent. In addition, despite it is negative result, the data for reviewer 3 question 2 should be included in the manuscript.

Thanks for the reviewer's suggestion. We have added the figure of RAD51 foci with or without POIII inhibitors and siRNase H1 treatment in new **Supplementary Figure 3C**. Notably, both PolIII inhibitors and siRNaseH 1 treatment attenuated the SYCP2 OE-induced RAD51 foci after damage, supporting the notion that SYCP2-RAD51-mediated repair is transcription and R-loop-dependent. We regret the oversight in not including and explaining this figure in the initial version. In the updated manuscript, we added it into figure and discussed the results.